# Synergistic and antagonistic activities of *IRF8* and *FOS* enhancer pairs during an immune-cell fate switch

Antonios Klonizakis [1,2], Marc Alcoverro-Bertran[1,3,7], Pere Massó[1,7], Joanna Thomas [1,4], Luisa de Andrés-Aguayo[1], Xiao Wei [1,2], Vassiliki Varamogianni-Mamatsi[5], Christoforos Nikolaou [6✉] & Thomas Graf [1,2✉]

## Abstract

Cell fate instructive genes tend to be regulated by large clusters of enhancers. Whether and how individual enhancers within such clusters cooperate in regulating gene expression is poorly understood. We have previously developed a computational method, SEGCOND, which identifies hubs that we termed Putative Transcriptional Condensates (PTCs), consisting of enhancer clusters and associated target genes. Here, we use SEGCOND to identify PTCs in a CEBPA-induced B-cell-to-macrophage transdifferentiation system. We find that PTCs are enriched for highly expressed, lineage-restricted genes and associate with BRD4, a component of transcriptional condensates. Further, we performed single and combinatorial deletions of enhancers within two PTCs active during induced transdifferentiation, harboring *IRF8* and *FOS*. Two enhancers within the *IRF8* PTC were found to provide a backup mechanism when combined, safeguarding *IRF8* expression and efficient transdifferentiation. Unexpectedly, two individual enhancers within the *FOS* PTC antagonize each other on day 1 of transdifferentiation, delaying the conversion of B-cells into macrophages and reducing *FOS* expression, while on day 7, they cooperate to increase *FOS* levels induced cells. Our results reveal complex, differentiation-stage-specific interactions between individual enhancers within enhancer clusters.

**Keywords** Enhancer Clusters; Transcription Factors; Cell Reprogramming; Enhancer Interactions; Enhancer Synergy
**Subject Categories** Chromatin, Transcription & Genomics; Haematology; Immunology

## Introduction

How cell-type-specific gene expression patterns are established during development and differentiation is a major research topic in current biology (Nora et al, 2023). Evidence accumulated over the past decades has revealed enhancers as major determinants of gene regulation and cell fate determination (Gasperini et al, 2020). Enhancers are sequence-specific DNA elements that recruit transcription factors and co-factors, and that interact with other enhancers as well as with promoters, increasing gene expression. It is well-accepted that large clusters of enhancer elements, described as "LCRs", "super" or "stretch" enhancers, regulate many cell fate-specifying genes and drive high levels of gene expression (Grosveld et al, 1987; Parker et al, 2013; Whyte et al, 2013). The role of individual enhancers within such large assemblies is not fully understood (Panigrahi and O'Malley, 2021).

Whether individual enhancers within clusters act independently or synergistically is an area of active research (Uyehara and Apostolou, 2023). Some studies suggested that they are essentially independent and act in an additive manner, ensuring that after the loss of one or more enhancers, genes critical for differentiation remain activated (Bahr et al, 2018; Osterwalder et al, 2018). According to this model, individual enhancers in the cluster may be completely or partially redundant, showing only mild effects when ablated individually. Thus, enhancer clustering could act as a "safeguard" mechanism by conferring expression robustness to their target genes. Other studies, however, have argued that the coalescence of multiple individual enhancers in an assembly may lead to synergistic activatory effects and, consequently, very high expression levels (Hnisz et al, 2017). Indeed, synergistic interactions among individual enhancer elements have been described (Blayney et al, 2023; Thomas et al, 2021; Choi et al, 2021). Enhancer synergy has been proposed to be achieved via the formation of transcriptional condensates (Hnisz et al, 2017). Such condensates are membrane-less organelles that contain multiple transcription factors and co-factors, such as MED1 and BRD4, and are described

[1]Genome Biology Program, Centre for Genomic Regulation, The Barcelona Institute of Science and Technology, 08003 Barcelona, Spain. [2]Department of Medicine and Life Sciences, Universitat Pompeu Fabra, 08005 Barcelona, Spain. [3]Josep Carreras Leukaemia Research Institute (IJC), 08916 Badalona, Spain. [4]Faculty of Life Sciences and Medicine, King's College, WC2R 2LS London, UK. [5]Institute of Pathology, University Medical Center Göttingen, 37075 Göttingen, Germany. [6]Institute for Bioinnovation, Biomedical Sciences Research Centre "Alexander Fleming", 16672 Vari, Greece. [7]These authors contributed equally: Marc Alcoverro-Bertran, Pere Massó. ✉E-mail: cnikolaou@fleming.gr; thomas.graf@crg.eu

to assemble at super-enhancers (Sabari et al, 2018). However, the role of transcriptional condensates is still somewhat controversial (Stortz et al, 2024).

Several computational frameworks have been developed for the genome-wide identification of enhancer clusters (Murphy et al, 2023; Klonizakis et al, 2023; Mota-Gómez et al, 2022; Parker et al, 2013; Whyte et al, 2013). One of the most widely used is the ROSE algorithm, which detects enhancer clusters termed super-enhancers (Whyte et al, 2013). ROSE stitches together a priori identified enhancers and ranks them based on the occupancy of a master transcription factor or a transcriptional co-activator such as MED1 or BRD4. Based on such single ChIP-seq datasets, enhancers that surpass an occupancy threshold are classified as super-enhancers. To improve the detection of enhancer clusters and their target genes, we have previously developed a computational algorithm, SEGCOND, which integrates multiple distinct datasets, including 3D genome conformation data (Klonizakis et al, 2023).

The SEGCOND algorithm partitions the genome into segments based on chromatin accessibility, histone mark, and transcription factor occupancy data. It then pinpoints segments that are enriched for H3K27ac-decorated enhancer elements and, via the integration of Hi-C data, selects those that contain strongly interacting enhancer–enhancer or enhancer–gene pairs. As a final step, SEGCOND links together enhancer-enriched segments of the same chromosome that associate in 3D space (Klonizakis et al, 2023). The final identified regions were originally coined as "Putative Transcriptional Condensates" (PTCs), based on the hypothesis that enhancer clusters serve as scaffolds for the formation of transcriptional condensates (Sabari et al, 2018; Hnisz et al, 2017). PTCs can thus consist of one, or more genomic segments and contain enhancers and their target genes. PTCs partially overlap with super-enhancers, with about 20% of super-enhancers falling within PTCs. However, additional uniquely PTC-associated genes are highly expressed, more so than super-enhancer unique genes (Klonizakis et al, 2023). We have applied SEGCOND to datasets derived from three timepoints of a CEBPA-induced B-cell-to-macrophage transdifferentiation system (Rapino et al, 2013). The system represents a highly efficient and homogenous differentiation process, enabling the study of enhancer clusters in a cell fate conversion context.

Here, we set out to investigate how individual enhancers within selected PTCs affect B-cell-to-macrophage transdifferentiation and the levels of target gene expression. Studying the role of individual enhancers in PTCs associated with IRF8 and FOS, using single and combined enhancer excisions, revealed additive, synergistic, and even antagonistic interactions. Our findings also indicate that the enhancer clusters studied behave distinctly in different stages of the immune cell fate conversion, suggesting complex regulatory modes that are likely shaping the differentiation process itself.

## Results

### PTCs dynamically form and dissociate during an immune cell fate switch

To decipher how large enhancer assemblies act during a cell fate conversion process we used a human B-ALL cell line (BLAER) containing an estrogen (E2) inducible CEBPA construct whose activation results in macrophage transdifferentiation within 7 days

(Rapino et al, 2013) (Fig. 1A). We previously identified three-dimensional enhancer clusters at three timepoints during transdifferentiation (Day 0, Day 1 and Day 7), employing SEGCOND (Klonizakis et al, 2023), identifying 333, 373, and 271 PTCs at Days 0, 1, and 7, respectively. Each PTC consisted of one, or more, genomic regions whose median sizes ranged from 275 kb to 292.5 kb (Fig. EV1A) and often contained multiple genes. After filtering for expressed genes ("Methods"), the median numbers of genes found in PTCs at the three differentiation stages were 6, 5, and 6 (Fig. EV1B). We also assessed the number of called H3K27ac peaks per PTC and found a median of 10–13 H3K27ac peaks at the three timepoints (Fig. EV1C).

To analyze the properties of genes associated with our identified PTCs at the early and late stages of transdifferentiation we clustered genes into seven groups according to the timepoint at which they were found to be linked with a PTC (Fig. 1B). Of 4524 genes identified in at least one timepoint, 1453 genes were found within a PTC throughout the whole process ("Invariant"). In contrast, the majority of genes (3071) exhibited a dynamic behavior, with 520 being associated with a PTC only at Day 0 ("Early"), 589 at Day 1 ("Intermediate"), and 700 at Day 7 "Late". Moreover, 661 genes were found in PTCs at both Day 0 and Day 1 ("Early-Intermediate"), while 339 were shared in Day 1 and Day 7 PTCs ("Intermediate-Late"). These data show that the association of genes with PTCs during transdifferentiation is a stage-dependent, dynamic process (Fig. 1B).

We next determined whether the genes associated with PTCs in different clusters correlate with the cells' differentiation state. At the early stages of transdifferentiation, cells adopt a B-cell identity, which is replaced by a myeloid identity as transdifferentiation progresses. Indeed, "Early-Intermediate" membership correlates with B-cell functions, while "Intermediate-Late" and "Late" correlate with myeloid functions (Fig. EV1D). Several B-cell identity genes such as IKZF1 (Heizmann et al, 2013), CD19 (Wang et al, 2012), and EBF1 (Hagman et al, 2012) are members of both Day 0 and Day 1 PTCs, while myeloid-related genes such as IRF8 (Tamura et al, 2000), FOS (Cai et al, 2007), CEBPB (Friedman, 2007), and CSF1R (Dai et al, 2002) are found in PTCs at Day 1 and/or Day 7 of transdifferentiation (Fig. 1C). Finally, cell cycle regulation GO terms are overrepresented at different stages, reflecting the fact that BLAER cells become quiescent towards their transdifferentiation to macrophages (Rapino et al, 2013) (Fig. EV1D).

We have previously shown that PTC-associated genes are highly expressed (Klonizakis et al, 2023). However, the way the expression of PTC-associated genes varies over time and whether it reflects the kinetics of B-cell-to-macrophage conversion is still unclear. We therefore examined expression levels of genes within PTC groups, using their Day 0 values as a reference. This showed that the expression levels of genes associated with PTCs in early stages of transdifferentiation ("Early", "Early-Intermediate") decrease after induction, while those of genes associated with PTCs at 1 or 7 days after induction ('Intermediate-Late' or 'Late') increase, as expected (Figs. 1D and EV1E). In some instances, we noticed only moderate changes in gene expression (Fig. EV1E). Given the large size of PTCs, we cannot exclude that only a specific set of genes within each PTC is predominantly affected by changes in chromatin structure and three-dimensional chromatin interactions. Thus, CTCF binding and/or promoter–enhancer compatibility could be responsible for local insulation effects within PTCs, restricting expression changes to a subset of genes.

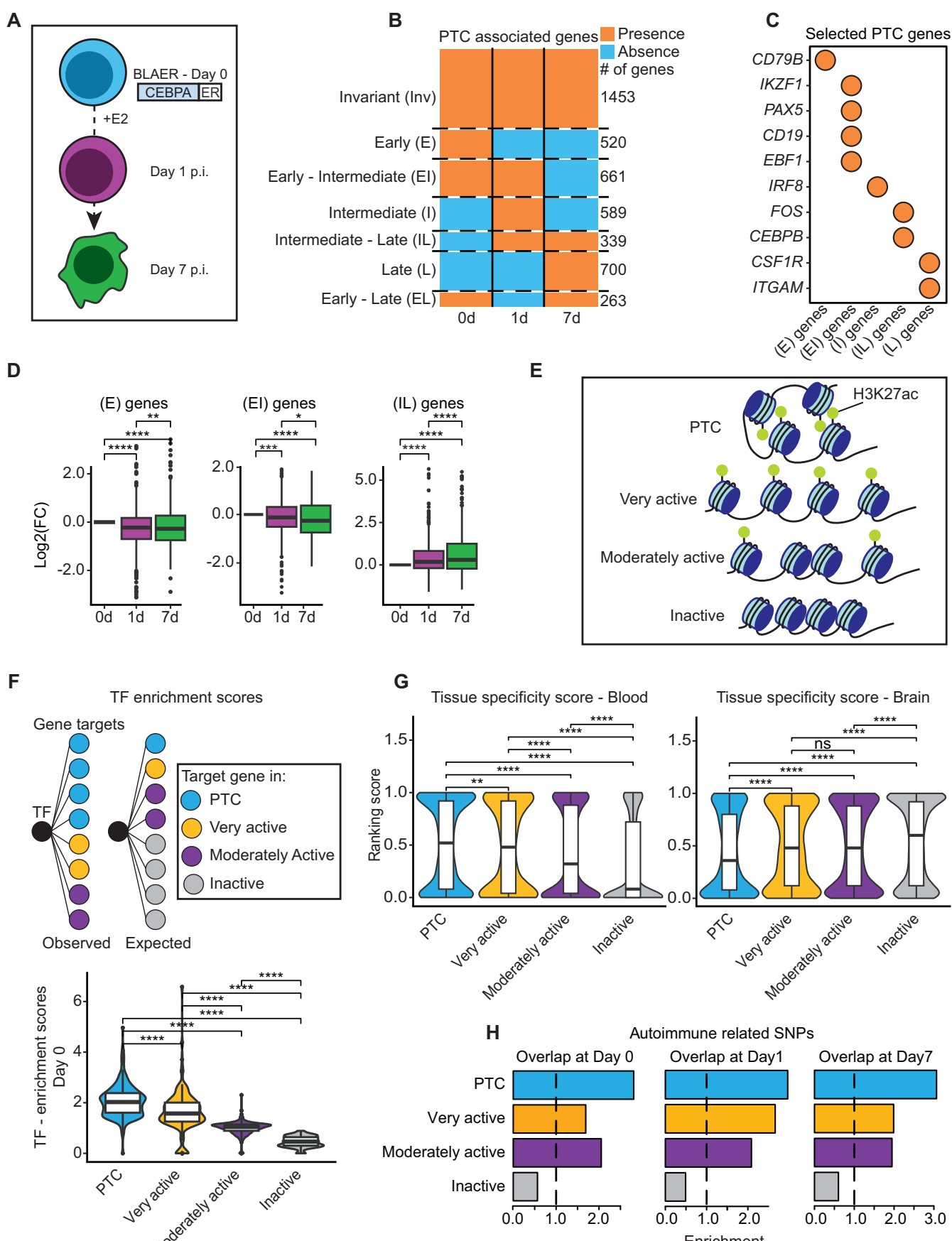

**Figure 1. SEGCOND–identified Putative Transcriptional Condensates (PTCs) are dynamic and harbor highly expressed, specialized genes.**

(A) Overview of CEBPA-induced B-cell-to-macrophage transdifferentiation. BLAER cells, a B-cell ALL cell line derivative, express CEBPA fused with the estrogen receptor (ER). Upon the addition of E2 (b-estradiol), cells transdifferentiate into macrophages over seven days (Rapino et al, 2013). PTCs were previously identified for Day 0, Day 1, and Day 7 cells (Klonizakis et al, 2023). (B) Clustering of filtered PTC-associated genes. Expressed protein-coding genes, with an H3K27ac-decorated promoter, were clustered according to the timepoint they were associated with a PTC in seven different groups. The PTC clusters were termed "Invariant" (Inv), "Early" (E), "Early-Intermediate" (EI), "Intermediate" (I), "Intermediate -Late" (IL), "Late" (L) and "Early-Late" (EL). (C) B-cell and macrophage-specific genes and their association with PTC clusters. Circles indicate gene presence within the corresponding cluster. (D) Expression dynamics of filtered PTC-associated genes. DESEq2 variance stabilized counts (Love et al, 2014) of genes grouped in (B) are depicted for three of the seven clusters. Values of each gene are normalized to their Day 0 expression value. Genes with very low DESEq2 variance stabilized counts at some timepoint were excluded from the analysis, as the normalized score could not be calculated. The paired Wilcoxon signed-rank test was used to determine statistically significant differences (*$P$ value ≤0.05; **$P$ value ≤ 0.01; ***$P$ value ≤ 0.001; ****$P$ value ≤ 0.0001). $N$ = 509 (E genes), 656 (EI genes), and 333 (IL genes). (E) Separation of the genome into four distinct groups via SEGCOND. Briefly, SEGCOND first partitions the genome into large segments, using ATAC-seq, H3K27ac, H3K4me3, and CEBPA ChIP-seq datasets. Segments are then scored for the presence of H3K27ac-decorated, accessible sites (enhancers) with a $P$ value and a log2FC value. $P$ values of ≤0.05 indicate enrichment or depletion of enhancer sites compared to random, background genomic regions, while log2FC values indicate whether segments contain more, or less, enhancers than expected by chance. Enhancer-enriched segments ($P$ value ≤0.05) that interact strongly in three-dimensional space, as determined by Hi-C data, are deemed as "PTC" segments. Enhancer-enriched segments that do not form 3D hubs are termed "Very Active" segments whereas segments that show no enhancer enrichment ($P$ value > 0.05) but a positive log2FC score are termed "Moderately active" segments. Finally, repressed segments are depleted of enhancer elements ($P$ value > 0.05 & log2FC ≤0). (F) Transcription factor–target enrichment scores for the four SEGCOND genomic groups at Day 0 of transdifferentiation. Transcription factor–target pairs were downloaded by the hTFtarget database (Zhang et al, 2020). An enrichment score, for each transcription factor, was used to determine whether its target genes are overrepresented in any of the SEGCOND genomic groups. Statistically significant differences were determined via a paired Wilcoxon rank-sum test (****$P$ value ≤0.0001). $N$ = 131 for all genomic region types. (G) Blood and brain tissue-specificity scores of all protein-coding genes associated with SEGCOND genomic categories at Day 0 cells. GTEx data were utilized to assign two scores to each gene, in the 0–1 interval, indicating whether the gene is preferentially expressed in blood or brain tissues respectively (see "Methods" for details). Statistically significant differences were determined via a Wilcoxon rank-sum test (ns $P$ value > 0.05; **$P$ value ≤0.01; ****$P$ value ≤0.0001). $N$ = 3668 (PTC), 1394 (Very active), 5942 (Moderately active), 8170 (Inactive). (H) Overlap enrichment of SEGCOND genomic categories with single nucleotide polymorphisms (SNPs) associated with autoimmune disorders. Enrichment was determined based on a permutation analysis. The dashed line corresponds to the baseline enrichment value of 1, indicating no significant overlap enrichment. Source data are available online for this figure.

In conclusion, a large proportion of PTCs exhibit a dynamic behavior during the induced B-cell-to- macrophage transdifferentiation and correlate with the expression levels of lineage-specific genes involved in the process.

## Genes associated with immune cell PTCs show complex regulation, tissue preference, and disease associations

PTCs are enriched for lineage-restricted, highly expressed genes (Klonizakis et al, 2023). We set out to further investigate the properties of these genes by comparing them with other expressed genes in our system that are not under the same multi-enhancer control. To do so, we used SEGCOND to separate the genome into four categories based on chromatin accessibility (ATAC-seq), H3K27ac decoration, and genome topology (Hi-C) (Fig. 1E). "PTCs" are three-dimensional enhancer clusters identified by SEGCOND (5.5% of the genome), "Very Active" regions are enriched in enhancer elements but do not form three-dimensional hubs (2.5% of the genome), "Moderately Active" are regions neither enriched nor depleted from enhancer elements that also harbor active genes (17.8% of the genome) and "Inactive" regions are depleted from enhancer elements and mainly harbor not expressed genes (74.2% of the genome) (Fig. 1E, see "Methods" for more details and Appendix 1 for examples).

Whether enhancer hubs in the eukaryotic nucleus are established through the association of multiple, diverse transcription factors or a small set of master transcriptional regulators is an open question. To explore this, we identified transcription factors (TFs) that are expressed in our cell system and matched them to their target genes, utilizing information from the hTFtarget database (Zhang et al, 2020). We then classified the target genes of each TF by their localization in the genome and assigned them to one of the genomic categories defined by SEGCOND (Fig. 1F). We next calculated a "background" network that randomly assigned the

target genes to genomic categories according to their total sizes. Using this approach, we calculated an enrichment score for each SEGCOND-defined genomic category and each transcription factor, quantifying whether or not target genes of a given TF are overrepresented. We found that PTCs are enriched for target genes of multiple distinct TFs compared to the other categories at all transdifferentiation stages (Figs. 1F and EV1F). These results indicate that PTC-associated genes are targeted by a larger variety of transcription factors than other genomic regions containing active genes.

The complex regulation of PTC-associated genes suggests that they are highly specialized, being mostly active in selected cell types. To further address this possibility, we utilized datasets from the GTEx database (Lonsdale et al, 2013) using as controls genes within "Very Active" "Moderately Active" and "Inactive" regions. To quantify how specialized the expression pattern of a given gene is, we created a custom "blood" tissue-specificity score. The score ranges from 0 to 1 and describes whether a gene's expression is higher in the GTEx blood samples than in 25 other types of tissues. A score of 1 indicates that a gene's expression is highest in blood samples, while 0 indicates it is at its lowest. Using this score, we observed significant enrichment for PTCs to contain genes that are preferentially expressed at higher levels in blood tissues, more so than genes in the "Very" and "Moderately Active" regions (Fig. 1G). We then repeated the analysis for "brain" related GTEx samples and observed significant differences between the clusters but in the opposite direction, i.e., the lowest association with PTCs (Fig. 1G). These observations indicate that PTC-associated genes identified in our transdifferentiation system are predominantly active in blood cell types.

The observation that PTCs are associated with a set of specialized genes active in blood tissues raised the possibility that they are linked to blood-related diseases, such as autoimmunity. To this end, we isolated autoimmune-related SNPs from a published study (Maurano et al, 2012) and calculated an enrichment score,

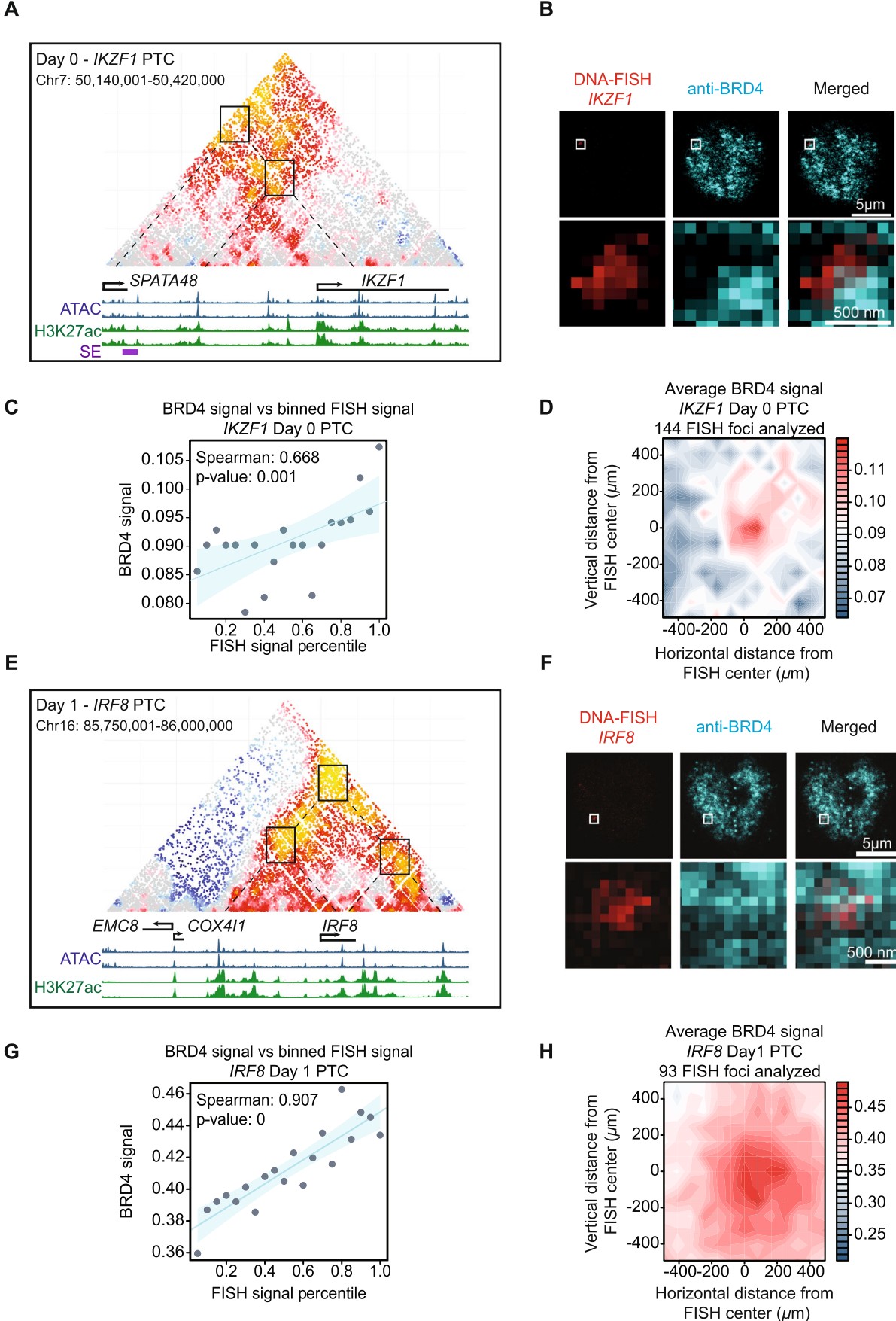

**A**
Day 0 - *IKZF1* PTC
Chr7: 50,140,001-50,420,000

SPATA48    IKZF1
ATAC
H3K27ac
SE

**B**
DNA-FISH *IKZF1*    anti-BRD4    Merged
5µm
500 nm

**C**
BRD4 signal vs binned FISH signal
*IKZF1* Day 0 PTC
Spearman: 0.668
p-value: 0.001
BRD4 signal
FISH signal percentile

**D**
Average BRD4 signal
*IKZF1* Day 0 PTC
144 FISH foci analyzed
Vertical distance from FISH center (µm)
Horizontal distance from FISH center (µm)

**E**
Day 1 - *IRF8* PTC
Chr16: 85,750,001-86,000,000
EMC8    COX4I1    IRF8
ATAC
H3K27ac

**F**
DNA-FISH *IRF8*    anti-BRD4    Merged
5µm
500 nm

**G**
BRD4 signal vs binned FISH signal
*IRF8* Day 1 PTC
Spearman: 0.907
p-value: 0
BRD4 signal
FISH signal percentile

**H**
Average BRD4 signal
*IRF8* Day1 PTC
93 FISH foci analyzed
Vertical distance from FISH center (µm)
Horizontal distance from FISH center (µm)

◀

**Figure 2.   The transcriptional condensate component, BRD4, overlaps with selected PTCs.**

(A) Overview of the *IKZF1* PTC at Day 0 non-induced cells. SHAMAN (Mendelson Cohen et al, 2017) Hi-C normalized values are depicted, with red and yellow points indicating contact enrichment between loci. ATAC-seq and H3K27ac ChIP-seq tracks of two biological replicates are also illustrated. Boxes correspond to H3K27ac-decorated elements that interact strongly (dashed lines) in 3D space. A super-enhancer (SE, purple box) within the PTC is also identified by the ROSE algorithm (Whyte et al, 2013). (B) DNA-FISH coupled with BRD4 immunofluorescence, targeting the *IKZF1* PTC at Day 0 non-induced cells. BRD4 signal (in cyan) overlaps with the *IKZF1* PTC (in red). The second row of images corresponds to the zoomed region falling within the highlighted area (white frame). The FISH probe spans 170 kb of the PTC. (C) DNA-FISH and BRD4 signal correlation analysis of Day 0 *IKZF1* PTC data. DNA-FISH pixel intensity values around identified FISH spots were binned in twenty percentiles. Matching BRD4 intensity values from the same pixels were assigned to each bin with the median of all BRD4 values depicted. (D) Contour plots of BRD4 signal enrichment over *IKZF1* Day 0 DNA-FISH centers. DNA-FISH spot centers were identified as described in "Methods". A window of 11 × 11 pixels was centered at each FISH spot and BRD4 intensity values were extracted for each position and each FISH spot. The median BRD4 intensity values, across all DNA-FISH spots, were calculated for each position of the 11 × 11 window. The process was repeated for randomly allocated FISH centers. Red/blue color codes correspond to the relative enrichment of BRD4 intensity compared to randomized FISH spots. (E) Overview of the *IRF8* PTC on Day 1 of transdifferentiation. Same as in (A). ROSE does not detect a super-enhancer within the PTC. The FISH probe spans 160 kb of the PTC. (F) DNA-FISH coupled with BRD4 immunofluorescence, targeting the *IRF8* PTC at Day 1 cells. Same as in (B). (G) DNA-FISH and BRD4 signal correlation analysis for Day-1 *IRF8* PTC data. Same as in (C). (H) Contour plots of BRD4 signal enrichment over *IRF8* Day 1 DNA-FISH centers. Same as in (D).

using a permutation approach, for all four genomic categories (see "Methods"). The results show that autoimmune-related SNPs are strongly enriched in PTCs across all timepoints (Fig. 1H), compared to the other genomic regions that also harbor active genes. To determine whether this tendency is specific to autoimmune-related SNPs, we repeated the same analysis using SNPs associated with neurological and behavioral traits and observed no enrichment within PTCs (Fig. EV1G). We also tested whether PTC genes at the early transdifferentiation stages are related to cancer. Indeed, genes in Day 0 and Day 1 PTCs are enriched in the "Lymphoma" category of the Human Phenotype Ontology atlas ($q$ values of 0.01502 and 0.0009) as assessed by the Enrichr software (Kuleshov et al, 2016). Of note, Day 7 PTC genes are not significantly enriched, likely because transdifferentiated macrophages exhibit a more quiescent behavior.

In conclusion, using our immune cell system, SEGCOND identified a set of specialized genomic regions that are under the regulatory control of multiple distinct TFs and that are associated with genes involved in autoimmune diseases.

## The transcriptional condensate-associated component BRD4 overlaps with selected PTCs

BRD4 is a protein component of transcriptional condensates that has been described to co-localize with super-enhancers based on DNA-FISH and immunofluorescence experiments (Sabari et al, 2018). An interesting question therefore is whether PTCs are associated with transcriptional condensates. We therefore investigated whether BRD4 associates with PTCs in our system, monitoring its presence in the *IKZF1* and *IRF8* associated PTCs, as examples representing a B cell and a macrophage PTC, respectively (Heizmann et al, 2013; Tamura et al, 2000).

We first focused on the 50.14–50.42 Mb PTC of *IKZF1* on chromosome 7, as identified by Hi-C, ATAC-seq, and H3K27ac ChIP-seq (Fig. 2A). This PTC is active at both Day 0 and Day 1 of transdifferentiation and overlaps with a super-enhancer (Fig. 2A). Using fixed samples of uninduced BLAER cells we marked the *IKZF1* PTC with a DNA-FISH probe and then stained the cells with a BRD4 antibody. This showed that the sites labeled with the two methods co-localize (Fig. 2B). To evaluate whether the degree of the overlap is more than expected by chance, we developed a computational approach. It first determines whether the DNA-FISH signal correlates with the BRD4 signal on a fixed window

around the center of DNA-FISH spots. It then evaluates whether the BRD4 signal is higher in the DNA-FISH spots' centers than in randomly allocated spots around cell nuclei (see "Methods" for details). Our analysis revealed that the DNA-FISH signal of uninduced cells correlated significantly with the BRD4 signal (Fig. 2C) and that BRD4 is enriched on DNA-FISH centers when compared with randomly allocated centers (Fig. 2D). Similar results were obtained with Day 1-induced cells after targeting the same locus (Fig. EV2A–D).

We next studied the *IRF8* PTC on chromosome 16 which is only active at Day 1 of transdifferentiation (Fig. 2E). Although *Irf8* super-enhancers have been reported in murine cells (Liu et al, 2023b), the ROSE algorithm did not identify them within the *IRF8* PTC in our system (Fig. 2E). We first evaluated whether BRD4 hubs assemble on this region predicted by SEGCOND, and observed that they indeed overlap with the *IRF8* PTC in Day 1-induced cells (Fig. 2F). Applying our computational analysis pipeline to DNA-FISH intensity and BRD4 signal at this locus yielded a highly significant correlation at Day 1 of transdifferentiation (Fig. 2G), as well as an enrichment of BRD4 signals in *IRF8* DNA-FISH centers compared to randomly allocated spots (Fig. 2H). No such findings were made with Day 0 cells, as the PTC is not active at that stage (Fig. EV2E,F).

In conclusion, our results indicate that BRD4 hubs are present at PTCs that are specifically formed at either Day 0 and 1 (*IKZF1*) or Day 1 cells (*IRF8*), in line with the idea that they correspond to enhancer regions that form condensates.

## Two enhancers within the PTC of *IRF8* confer high *IRF8* expression levels and transdifferentiation robustness

To study how enhancers within PTCs modulate gene expression and transdifferentiation, we chose the *IRF8* PTC as a test case. *IRF8* is a known myeloid regulator (Tamura et al, 2000) which is upregulated approximately 16-fold in our transdifferentiation system at Day 1 post-CEBPA induction. The *IRF8* PTC contains multiple individual enhancers, and their deletion in all possible combinations would be technically impractical. To prioritize enhancers for excisions, we inspected the chromatin profile of the region and the binding profiles of CEBPA and its obligate partner, PU.1 (Fig. 3A) (Van Oevelen et al, 2015; Heinz et al, 2010). We aimed to target enhancers that are not active in Day 0 cells and are co-bound by the two factors. We selected the only two enhancers meeting these criteria, located at −69 kb and +83 kb from the *IRF8*

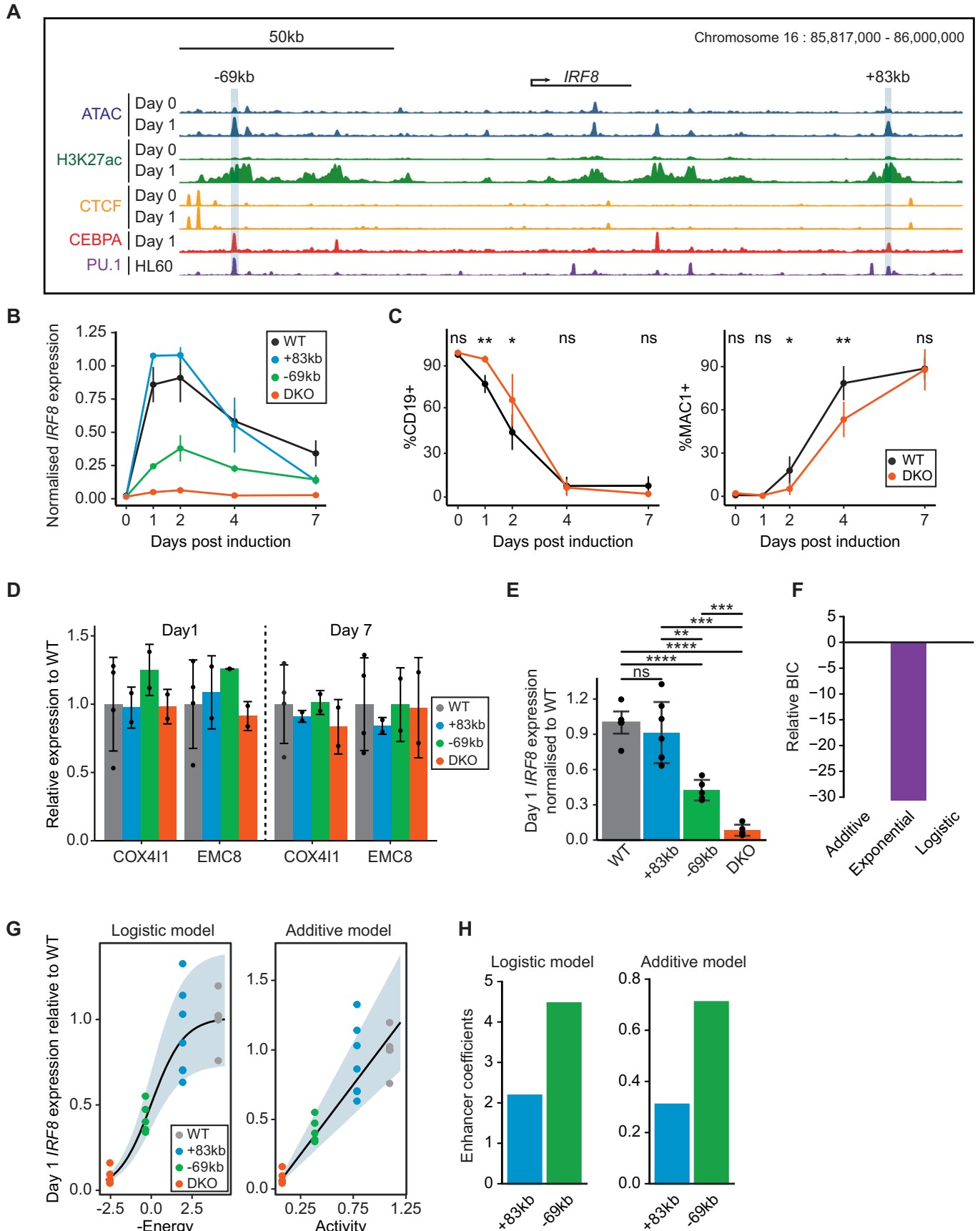

**Figure 3. The -69kb and + 83 kb *IRF8* enhancers provide robustness to *IRF8* expression levels and transdifferentiation.**

(A) Overview of the *IRF8* PTC chromatin landscape at Day 0 and Day 1 cells. ATAC-seq, H3K27ac, and CTCF ChIP-seq tracks of Day 0 and Day 1 cells are depicted. Two enhancers, −69 kb and +83 kb away from the *IRF8* transcription start site were chosen for excision, as they show an increase in H3K27ac decoration and chromatin accessibility at Day 1 cells, are not bound by CTCF and are bound by C/EBPa in Day 1 cells and PU.1 in HL60 cells. (B) *IRF8* expression kinetics during transdifferentiation. *IRF8* expression levels were monitored via RT-qPCR and were normalized against *GUSB* expression levels for each timepoint. The mean ± s.d. is depicted for each timepoint. $N = 4$ (WT Days 0, 2, 4,7), 3 (WT Day 1, −69 kb all timepoints), and 2 (+ 83 kb, DKO, all timepoints) biological replicates. (C) Transdifferentiation kinetics of *IRF8* DKO cells compared to BLAER cells, monitored via FACS. CD19+ and MAC1+ cells were evaluated across 5 timepoints. The mean ± s.d. is depicted for each timepoint. Statistical significance for each timepoint was determined using a two-way ANOVA test, factoring in transdifferentiation batches (ns $P$ value > 0.05; *$P$ value ≤0.05; **$P$ value ≤0.01). $N = 4$ (WT Days 0, 1, 2), 3 (DKO Days 0, 1, 2), and 2 (WT Days 4, 7 and DKO Days 4, 7) biological replicates. (D) Day 1 and Day 7 expression levels of other genes within the *IRF8* PTC. *COX4I1* and *EMC8* expression levels were quantified via RT-qPCR and normalized against *GUSB* expression. The normalized expression values were then divided by the average expression of WT cells. The mean ± s.d. are depicted for each gene, sample, and timepoint. Individual measurements are also depicted as points. Statistical significance of expression differences between cell lines was determined using the Student's $t$ test and no significant hits were found. $N = 4$ (WT) and 2 (−69 kb, 83 kb, DKO) biological replicates. (E) Day 1 *IRF8* expression levels of all generated KO lines and WT cells. *IRF8* expression levels were determined via RT-qPCR and normalized against *GUSB* expression levels. Consequently, expression values from each transdifferentiation batch were normalized to the average WT expression values of the same batch. The mean ± s.d. is depicted, alongside individual measurements as points. Statistical significance was determined using a Student's $t$ test (ns $P$ value > 0.05; **$P$ value ≤0.01; ***$P$ value ≤0.001; ****$P$ value ≤0.0001). $N = 12$ (WT), 7 (+ 83 kb), and 5 (−69 kb, DKO) biological replicates. (F) Evaluation of best statistical model fit for Day 1 *IRF8* expression data. The Bayesian Information Criterion (BIC) was used to determine which model better fits the expression data. BIC values of the additive loci were used as a reference point. A BIC difference greater than 2 indicates a better fit. Both the additive and logistic models were found to be valid. (G) Logistic and additive model fits to *IRF8* Day 1 expression data. Black lines denote the models' average predicted expression values for each enhancer combination, with the shaded area corresponding to 0.1–0.9 quantiles. Individual expression values, normalized to WT values, are depicted as points. The x axis of the logistic model corresponds to the energy being conferred towards increasing transcriptional output by the addition of each enhancer. The x axis of the additive model corresponds to the activity of each enhancer which is proportional to gene expression. (H) Logistic and additive model enhancer coefficients for *IRF8* Day 1 expression data. The enhancer coefficients determine enhancer potency toward increasing expression levels. Both models showcase that the −69 kb enhancer is stronger than the +83 kb enhancer at this transdifferentiation stage. Source data are available online for this figure.

start site as they show an increase in their chromatin accessibility and H3K27ac decoration at Day 1 and are bound by both CEBPA and PU.1 (Fig. 3A). In addition, they are not bound by CTCF, eliminating a possible confounding parameter due to its role in genome topology (Ong and Corces, 2014) (Fig. 3A).

We proceeded in excising the −69 kb and +83 kb enhancers by CRISPR separately and in combination, producing three knockout (KO) lines. These lines were induced to transdifferentiate and monitored at five timepoints for the expression of *IRF8* via RT-qPCR. We expected that single enhancer excisions would result in a modest reduction of *IRF8* upregulation and that their combined excision would lead to a more severe reduction. However, surprisingly, while deletion of the +83 kb enhancer led to no obvious differences in *IRF8* levels at Days 1, 2, and 4 of transdifferentiation, excision of the −69 kb enhancer led to a strong *IRF8* RNA reduction, and the double knockout (DKO) to an even more severe decrease at all timepoints after induction (Fig. 3B).

We next measured by flow cytometry the protein levels of CD19 and MAC1 (CD11B) during transdifferentiation, permitting us to simultaneously assess the silencing of the B-cell program and activation of the macrophage program (Fig. EV3A). The results obtained showed that excision of the −69 kb enhancer slightly but significantly delayed CD19 downregulation on Days 1 and 2 without affecting MAC1 activation (Fig. EV3B). Excising the +83 kb enhancer likewise delayed B-cell silencing without affecting the macrophage program (Fig. EV3B). Strikingly, however, the double knockout (DKO) cells showed a significant delay not only in CD19 silencing but also in MAC1 activation (Fig. 3C). Of note, expression of these markers eventually converged in induced macrophages at Day 7 in all three knockout cell lines. We also tested whether the changes in transdifferentiation kinetics could be attributed to alterations in the expression of *COX4I1* and *EMC8* that fall within the *IRF8* PTC. However, the expression of the two genes at Day 1 and 7 KO cells is similar to WT cells, showing that the excised enhancers are *IRF8* specific (Fig. 3D).

The finding that the +83 kb enhancer KO caused a slight delay in B-cell marker repression, while it barely affected *IRF8* expression at Day 1 cells, was unexpected. To explore possible reasons, we performed additional RT-qPCR experiments to monitor *IRF8* expression in non-induced cells. This revealed that excision of the +83 kb enhancer reduced *IRF8* expression in Day 0 cells, suggesting that the enhancer is already active at this stage (Fig. EV3C). Although CEBPA induction substantially upregulates *IRF8* expression in Day 1 + 83 kb KO cells, we speculate that the observed slight delay in transdifferentiation velocity is due to the reduced expression levels of *IRF8* already in the cells prior to CEBPA induction.

In an attempt to dissect the possible interactions between the −69 kb and +83 kb enhancers in the *IRF8* PTC, we utilized three previously described statistical models, termed additive, exponential, and logistic (Dukler et al, 2017). Each model assumes a different relationship between the two enhancers, leading to different transcriptional outputs. According to the additive model, each enhancer independently and additively contributes to the strength of the enhancer cluster, while the exponential model suggests that enhancers within a cluster interact, amplifying or dampening each other's activity. Finally, the logistic model assumes that each enhancer contributes independently to the reduction of an energy threshold important for transcription (Fig. EV3D, see "Methods" for more details).

We fit the three models in our Day 1 expression data after including results from an additional independent cell clone for each condition. We observed that on Day 1 cells *IRF8* levels do not differ significantly between the wild type and the +83 kb *IRF8* KO cells (Fig. 3E). They are, however, reduced in -69kb cells, and further reduced in DKO cells (Fig. 3E). Both the logistic and additive models were found to be good fits for our Day 1 dataset, as determined by the Bayesian Information Criterion (BIC) (Fig. 3F). The logistic model predicts a sigmoid-like behavior for *IRF8* expression, where even small differences in enhancer activity can

lead to sharp reductions in expression levels, while the additive model suggests that the two enhancers' activity is added up linearly (Fig. 3G). Notably, both models highlight a large difference between the activity of the two enhancers, with the −69 kb enhancer being ~2.5 times stronger than the +83 kb enhancer at this stage of transdifferentiation (Fig. 3H). Finally, we also investigated whether our Day 1 *IRF8* expression measurements and statistical modeling were affected by the reduced *IRF8* levels in +83 kb and DKO uninduced cells. To do so, we captured nascent *IRF8* mRNA produced for 1 day after CEBPA induction and repeated our RT-qPCR experiments. We observed the same trend in *IRF8* expression levels (Fig. EV3E) and the logistic and additive models producing the best fits (Fig. EV3F) as in our bulk mRNA measurements. Thus, the initial drop in *IRF8* levels in Day 0 cells does not affect post-induction measurements and conclusions.

We then analyzed our *IRF8* expression dataset of Day 7 cells, at the induced macrophage stage. The deletion of either enhancer leads to a sharp reduction in *IRF8* expression, with their combined excision introducing an even stronger decrease (Fig. EV3G). The exponential model was the best fit, indicating functional synergy between the two enhancers (Figures EV3H,I). The model also suggests that the strength of the two enhancers was similar in Day 7 cells (Fig. EV3J), in contrast to their activities in Day 1 cells.

Overall, the two enhancers studied within the PTC of *IRF8* were found to control the gene's expression differently at distinct stages of transdifferentiation and to cooperate in driving high levels of *IRF8* expression. These effects translate into more complex outcomes at the level of transdifferentiation kinetics, suggesting thresholds of IRF8 important for both the silencing and activation of lineage markers.

## Enhancers within the *FOS* PTC show time-dependent negative and positive interactions during induced transdifferentiation

To determine whether the observed behavior of the two *IRF8* enhancers is a more general property of PTCs we analyzed the *FOS* locus, which is found within a PTC that forms at Days 1 and is maintained at Day 7 of transdifferentiation (Fig. EV4A). FOS has been reported to play a role in monocyte specification through dimerization with CEBPA (Cai et al, 2007). The *FOS* PTC spans 470 kb and contains multiple enhancers as well as other genes (Fig. EV4A). To assess whether BRD4 foci overlap with the PTC at Day 1 cells, we performed DNA-FISH experiments coupled with BRD4 immunofluorescence. Indeed, we observed a significant co-localization of BRD4 with the PTC (Fig. 4A–C).

Most H3K27ac-decorated regions within the PTC were similarly decorated in Day 0 and Day 1 cells (Fig. EV4A). In contrast, substantial chromatin changes occurred at two sites near the *FOS* gene on Day 1 (Fig. 4D). We thus focused on these −3.3 kb and +8.9 kb FOS proximal enhancers, which were bound by CEBPA but not by CTCF, and that were also captured by the super-enhancer algorithm (Fig. 4D). These enhancers were excised in the BLAER cell line both individually and in combination, generating three derivative lines.

We measured the expression levels of *FOS* across five timepoints after induction in the three cell lines (Fig. 4E). Unexpectedly, deletion of the −3.3 kb enhancer led to a transient upregulation of *FOS* at Days 1 and 2 of transdifferentiation, suggesting that at these timepoints the −3.3 kb enhancer functions as a repressor (Fig. 4E). This repression is not mediated by the polycomb complex as the −3.3 kb enhancer was

not found to be decorated with the repressive histone mark H3K27me3 in Day 1 cells (Fig. EV4B). In contrast, excision of the +8.9 kb enhancer led to a pronounced reduction in *FOS* expression at Day 1 and Day 1 cells, as seen for the DKO clones (Fig. 4E).

We next monitored the transdifferentiation dynamics of the knockout cells by FACS. Surprisingly, excision of the −3.3 kb enhancer substantially and significantly accelerated both the silencing of CD19 and activation of MAC1 (Fig. 4F). In contrast, ablation of the +8.9 kb enhancer did not alter the transdifferentiation kinetics (Fig. EV4C) while it counteracted the observed acceleration caused by the −3.3 kb enhancer excision in DKO cells (Fig. EV4C). These findings suggest that when *FOS* expression levels range between those observed in +8.9 kb/DKO cells and WT cells, transdifferentiation can proceed normally. Conversely, elevated levels of FOS, exceeding those in WT cells, accelerate it. We finally tested whether the observed alterations in transdifferentiation kinetics could be due to changes in *FOS* mRNA levels in non-induced knockout cells or due to the excised enhancers having other gene targets. However, *FOS* expression levels were not altered in Day 0 cells (Fig. EV4D), and the expression of *NEK9* and *TMED10* within the PTC was not changed (Fig. EV4E), ruling out these possibilities.

To exclude potential clone-specific biases, we generated a second set of cell clones for the two FOS enhancer knockouts and their combinations. Once again, we observed an increase of *FOS* expression at Day 1 −3.3 kb clones and a sharp decrease in the +8.9 kb and DKO clones (Fig. 4G). We then applied statistical modeling to our Day 1 expression data and determined that the exponential model results in the best fit (Fig. 4H). Under the assumption of this model, the two enhancer elements act synergistically to modulate each other's activity (Fig. 4I). However, in this case, the −3.3 kb enhancer displays a negative coefficient (Fig. 4J) as it antagonizes the activity of the +8.9 kb enhancer. Repeating the same analysis for Day 7 cells showed that the DKO cells, but not the individual KO cell lines, exhibited a decrease in *FOS* expression (Fig. EV4F). In contrast to Day 1 data, both the logistic and additive models were found to fit the Day 7 expression data better (Fig. EV4G,H). Under the latter models, the −3.3 kb enhancer now functions as an activator rather than as a repressor, although weaker than the +8.9 kb enhancer (Fig. EV4I). This result suggests that there is a switch in the behavior of the *FOS* PTC between Day 1 and 7 of transdifferentiation.

In conclusion, our data show that the −3.3 kb FOS enhancer antagonizes the activity of the +8.9 kb enhancer on Day 1, dampening FOS expression and delaying transdifferentiation. The antagonism was found to be Day 1 stage-specific, as the two enhancers cooperated in Day 7 cells, providing robustness to *FOS* expression levels.

## Enhancer excisions do not affect H3K27ac occupancy within PTCs

We went on to investigate the molecular mechanisms involved in the regulation of the *IRF8* and *FOS* genes by the studied enhancers. Enhancers could influence gene expression and transdifferentiation dynamics via a chromatin-structure-related mechanism. Thus, we speculated that enhancer excisions could affect H3K27ac deposition or chromatin accessibility in other, nearby regulatory regions of the same PTC. To explore this possibility, we first focused on the *IRF8* PTC and performed H3K27ac ChIP in *IRF8* −69 kb, +83 kb, and DKO cells at Day 1 of transdifferentiation. We then performed qPCR targeting

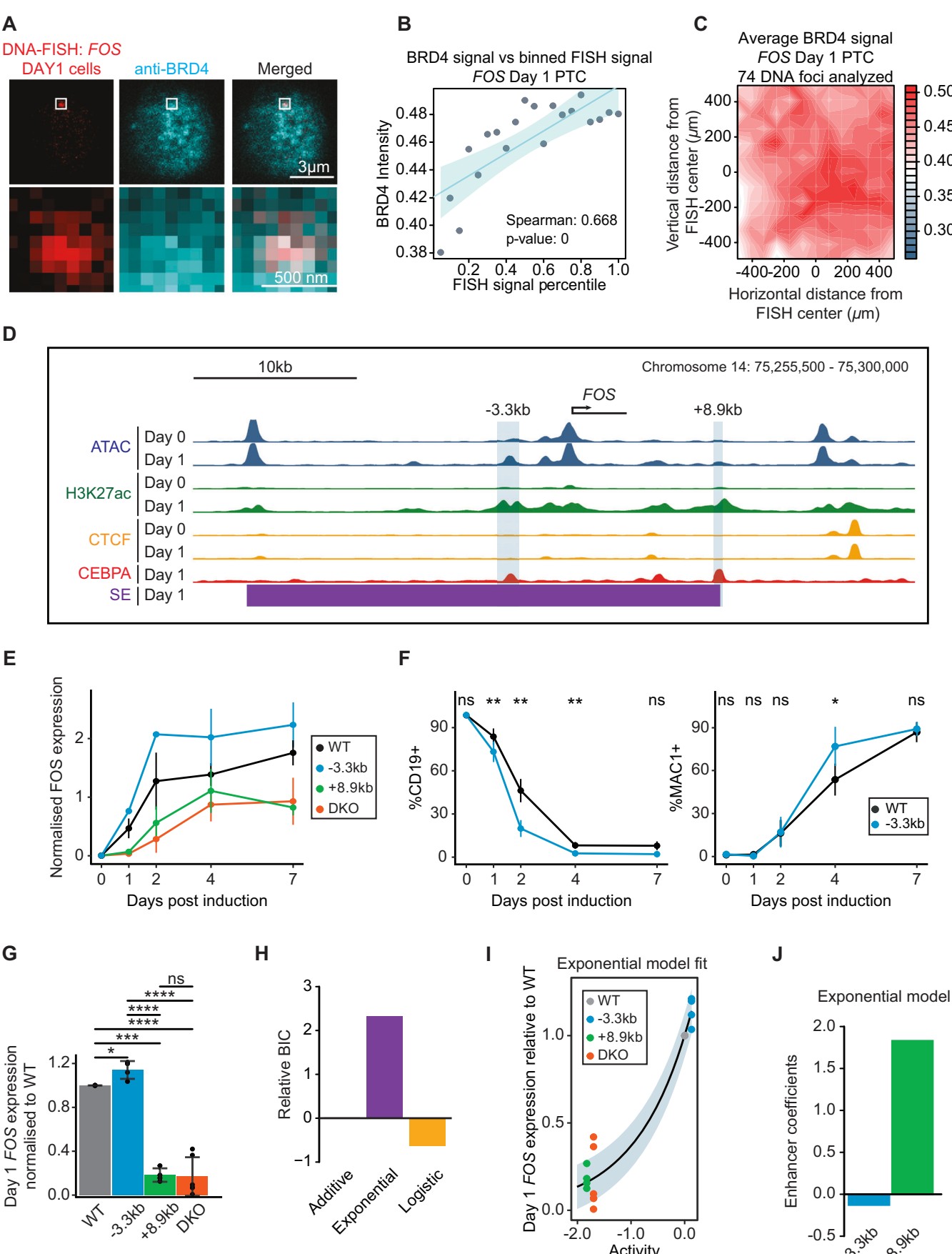

**Figure 4. The -3.3 kb and + 8.9 kb *FOS* enhancers antagonize at Day 1 cells, delaying transdifferentiation kinetics.**

(A) DNA-FISH coupled with BRD4 immunofluorescence, targeting the *FOS* PTC at Day 1 cells. BRD4 foci overlap with the *FOS* PTC. The FISH probe spans 236 kb of the PTC. (B) DNA-FISH and BRD4 signal correlation analysis of Day 1 *FOS* PTC data. Analysis as in Fig. 2C. (C) Contour plots of BRD4 signal enrichment over Day 1 *FOS* PTC DNA-FISH centers. Analysis as described in Fig. 2D. (D) Overview of part of the *FOS* PTC at Day 0 and Day 1 cells. ATAC-seq, H3K27ac, and CTCF ChIP-seq tracks of Day 0 and Day 1 cells are shown. Two enhancers, −3.3 kb and +8.9 kb from the *FOS* transcription start site, were chosen for excision, as they become more accessible, show an increase in H3K27ac signal at Day 1 cells, are not occupied by CTCF in any timepoint and are occupied by C/EBPa in Day 1 cells. A super-enhancer called by the ROSE algorithm is also identified in the *FOS* PTC, containing the two enhancer elements. (E) *FOS* expression levels throughout transdifferentiation. *FOS* expression levels were measured via qPCR and were normalized against *GUSB* expression levels. The mean ± s.d. is depicted for each timepoint. $N = 4$ (WT Days 0, 1, 2, 4), 3 (WT Day 7), and 2 (−3.3 kb, +8.9 kb, and DKO samples) biological replicates. (F) Transdifferentiation kinetics of *FOS* −3.3 kb KO compared to BLAER cells, assessed via FACS. The mean ± s.d. of CD19+ and MAC1+ cells is depicted for each timepoint. Statistical significance was determined for each timepoints using a two-way ANOVA test, factoring in transdifferentiation batches (ns $P$ value > 0.05; *$P$ value ≤0.05; **$P$ value ≤0.01). $N = 5$ (WT Days 0, 1, 2), 4 (WT Day 7, −3.3 kb Days 0, 1, 2, 7), and 3 (WT Day 4, −3.3 kb Day 4) biological replicates. (G) Day 1 *FOS* expression levels of KO and WT cells. *FOS* expression levels were measured via RT-qPCR and normalized to *GUSB* expression levels. Consequently, expression values from each transdifferentiation batch were normalized to the average WT expression values of the same batch. The mean ± s.d. is depicted, alongside individual measurements as points. Statistical significance was determined using a Student's $t$ test (ns $P$ value > 0.05; *$P$ value ≤0.05; ***$P$ value ≤0.001; ****$P$ value ≤0.0001). $N = 6$ (DKO), 5 (WT), and 4 (−3.3 kb and +8.9 kb) biological replicates. (H) Selection of best model fit for Day 1 *FOS* expression data. As in Fig. 3F. The exponential model was the better fit to the data. (I) Exponential model fit to *FOS* Day 1 expression data. Same as in Fig. 3G. (J) Exponential model enhancer coefficients for *FOS* Day 1 expression data. The −3.3 kb enhancer is assigned a negative coefficient as it dampens the activity of the +8.9 kb enhancer. Source data are available online for this figure.

three other H3K27ac peaks within the *IRF8* PTC, likely representing regulatory elements (Fig. 5A). A region devoid of H3K27ac was included as a negative control to assess ChIP quality. Our results indicate that the H3K27ac signal is not altered in the single KO or the DKO cell lines, suggesting that chromatin-associated changes cannot account for our findings with the *IRF8* PTC (Fig. 5A).

Similarly, the excision of the −3.3 kb *FOS* enhancer could repress the activity of the +8.9 kb enhancer via a chromatin-related mechanism. To test this, we performed H3K27ac ChIP experiments with WT and *FOS* −3.3 kb cells on Day 1 of transdifferentiation and targeted the +8.9 kb enhancer and a region 1 kb upstream of the *FOS* transcription start site. As observed in the case of the *IRF8* PTC, no changes were found to occur in the deposition of H3K27ac across the tested regions (Fig. EV5A) Together, our data show that H3K27ac histone decoration is not perturbed in any of the tested cellular backgrounds, ruling out the possibility that alterations of this enhancer mark drive gene expression changes.

## BRD4 foci associate with the IRF8 PTC in double knockout cells

An alternative explanation of how the excised enhancers affect gene expression is the involvement of BRD4. Since BRD4 foci associate with the *IRF8* and *FOS* PTCs in WT cells, the studied enhancers could recruit BRD4 and subsequently increase gene expression. To test this, we performed DNA-FISH experiments coupled with BRD4 immunofluorescence in *IRF8* DKO cells at Day 1 of transdifferentiation, where *IRF8* levels are reduced dramatically. Surprisingly, we observed that BRD4 foci still overlapped with the *IRF8* PTC (Fig. 5B). In order to evaluate whether the observed overlaps are more frequent than expected by chance, we used our computational pipeline to analyze our images. Indeed, the BRD4 signal strongly correlated with the DNA-FISH signal (Fig. 5C) and was enriched in DNA-FISH centers compared to randomly allocated FISH spots (Fig. 5D). We cannot rule out that the rest of the H3K27ac-decorated regions present in the *IRF8* PTC, which are not perturbed in DKO cells, are responsible for BRD4 recruitment. Nevertheless. these results suggest that BRD4 recruitment and *IRF8* expression are uncoupled and that changes in gene expression and transdifferentiation following the excision of *IRF8* PTC enhancers are not due to a defect in BRD4 recruitment.

## Deciphering transcription factor motif syntax in the IRF8 PTC

Disentangling the stage-specific and distinct behaviors of the PTC enhancers studied would require the identification of bound transcription factors. In an attempt to do this, we utilized a recently described algorithm, ChromBPnet (Pampari et al, 2024). ChromBPnet is a deep learning approach trained on ATAC-seq data that predicts chromatin accessibility profiles from the underlying DNA sequence. After training on ATAC-seq datasets, it assigns a "contribution score" that quantifies the effect of each base on the predicted chromatin accessibility profile. These scores can thus help determine DNA sequence motifs that are associated with an increase or decrease in chromatin accessibility and can be mapped to known transcription factor binding motifs using the JASPAR database. Since there is a high degree of similarity between motifs of factors corresponding to the same protein families, the transcription factor motif assignments described here remain to be experimentally validated.

We trained three different ChromBPnet models on our Day 0, 1, and 7 ATAC-seq data and calculated contribution scores for the excised enhancer regions within the *IRF8* PTC. Inspection of the -69kb enhancer revealed a set of motifs with high contribution scores, indicative of increased chromatin accessibility (Fig. 5E). Two of these motifs are present at all differentiation stages, while two appear only after CEBPA induction. By comparing the underlying sequences against the TF-motif JASPAR database, we discovered that the stable motifs resemble binding sites of the IKAROS and RUNX protein family (Fig. 5E). Members of the two families, such as *IKZF1* and *RUNX1*, are expressed at high levels throughout transdifferentiation (Fig. EV5B,C). The dynamic motifs are similar to PU.1 and CEBPA binding sites (Fig. 5E). *PU.1* is upregulated after CEBPA induction (Fig. EV5D) and collaborates with CEBPA to drive high expression of macrophage-related genes (Van Oevelen et al, 2015; Heinz et al, 2010). We speculate that the strength of the -69kb enhancer at this stage depends on the functional cooperation of the two factors. At Day 7 cells, both CEBPA and PU.1 contribution scores are reduced, which may reflect the diminished enhancer strength and *IRF8* expression at the induced macrophage stage.

A similar analysis of the +83 kb enhancer revealed that the motif syntax is "soft" with lower contribution scores assigned to each base (Fig. EV5E). ChromBPnet revealed EBF, RUNX, and TCF-related

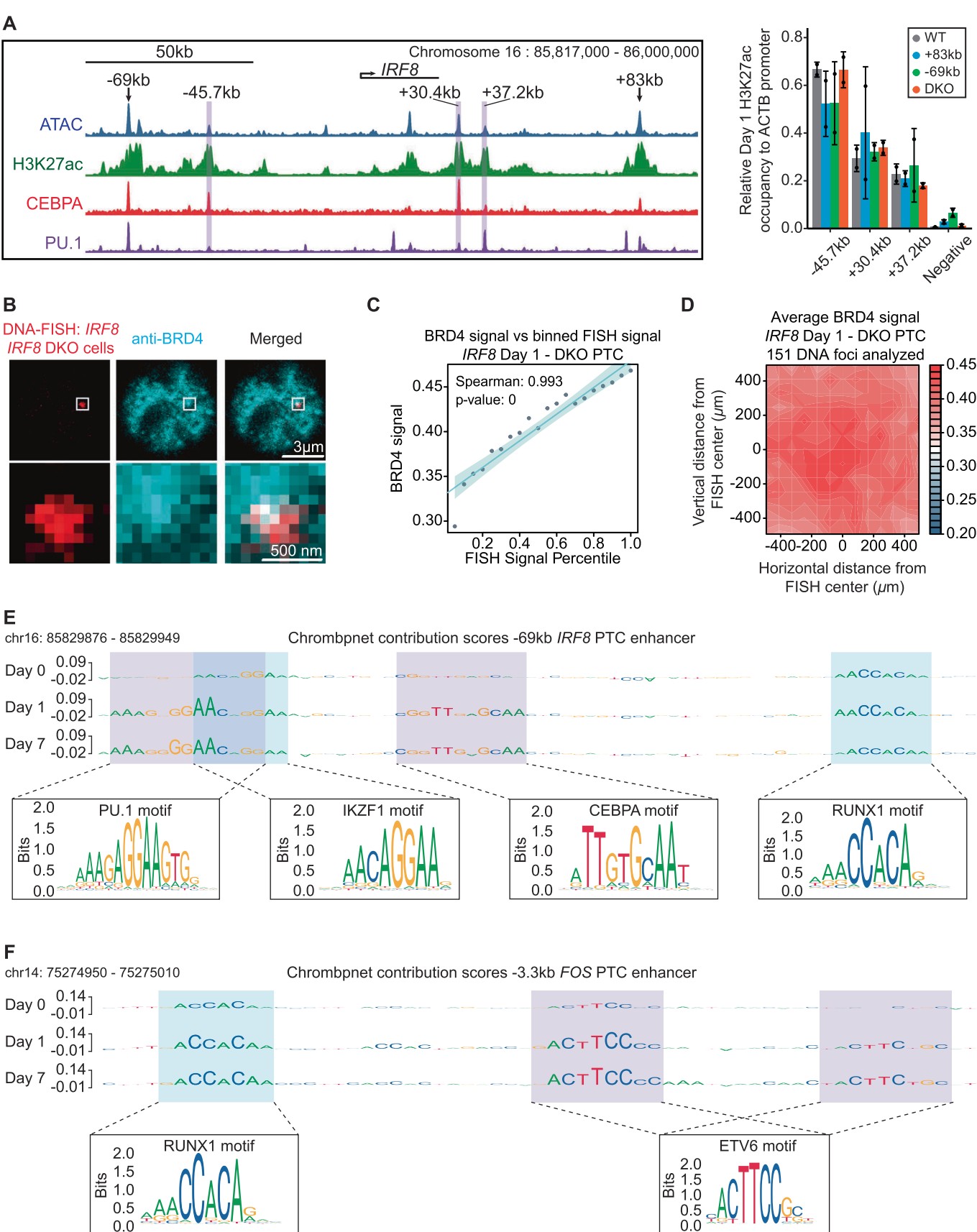

**Figure 5.  Deciphering the molecular events underlying PTC enhancer function.**

(A) H3K27ac ChIP-qPCR of *IRF8* PTC enhancers at Day 1 of transdifferentiation. Primers targeting three regions (highlighted in pink) within the *IRF8* PTC were used to determine the effects of enhancer excisions (indicated by arrows) in the H3K27ac signal across the *IRF8* PTC. ChIP-qPCR scores were calculated using the "Constant Amount" method described in (Solomon et al, 2021) and normalized to H3K27ac levels of the *ACTB* promoter. The mean ± s.d. are depicted for each region, with individual measurements included as points. A Student's *t* test was used to determine any significant differences in H3K27ac levels between cell lines. None of the comparisons was found to be statistically significant. N = 2 biological replicates for each cellular background and region. (B) DNA-FISH and BRD4 immunofluorescence, targeting the *IRF8* PTC at Day 1, DKO cells. BRD4 foci overlap with the PTC despite enhancer excision. (C) DNA-FISH and BRD4 signal correlation analysis of Day 1 *IRF8* PTC data in DKO cells. Analysis as in Fig. 2C. (D) Contour plots of BRD4 signal enrichment over Day 1 *IRF8* PTC DNA-FISH centers in DKO cells. Analysis as described in Fig. 2D. (E) ChromBPnet contribution scores in the −69 kb *IRF8* PTC enhancer at all transdifferentiation stages. A ChromBPnet model trained on ATAC-seq data can be used to assign contribution scores to each base pair of a region. Large contribution scores are assigned to bases that lead to increased chromatin accessibility predictions by the ChromBPnet model. Consecutive base pairs with high contribution scores were matched to known transcription factor binding motifs, using the JASPAR database. Matches are shown within frames. Motifs that are present in Day 0 cells are labeled in blue, while motifs that appear after CEBPA induction in Day 1 cells are labeled in pink. (F) ChromBPnet contribution scores across transdifferentiation in the −3.3 kb *FOS* PTC enhancer. Same as in (E). Source data are available online for this figure.

motifs at Day 0 and Day 1 cells (Fig. EV5E), transcription factors whose expression might be required to drive *IRF8* expression in non-induced cells. EBF1 and TCF3 are essential B-cell regulators (Nechanitzky et al, 2013; Bain et al, 1994), and their expression decreases after CEBPA induction (Fig. EV5F,G). The only motif appearing after CEBPA induction likely corresponds to AP1 family members, such as FOS, and could drive the enhancer's activity at Day 1 and 7 cells. The shift between the motif syntax observed on Day 1 and 7 cells may be responsible for the functional synergy of the two enhancers observed for Day 7 but not Day 1 cells.

In summary, the ChromBPnet analysis provided new insights into the dynamic behavior of the −69 kb and +83 kb enhancers during transdifferentiation.

## ChromBPnet predicts the presence of a repressor in the *FOS* −3.3 kb enhancer

Applying ChrombBPnet to the −3.3 kb *FOS* PTC enhancer revealed three motifs with high contribution scores (Fig. 5F). One motif is related to the RUNX family and is already present in Day 0 cells but shows increased scores in Day 1 and Day 7 cells. The two other motifs are highly similar to ETV6 binding sites and also show increased scores in Day 1 and Day 7 cells. ETV6, a transcriptional repressor associated with various types of leukemias (Hock and Shimamura, 2017), is transiently overexpressed in Day 1 cells, with its levels dropping again in Day 7 cells (Fig. EV5H). Thus, the observed antagonism between the -3.3 kb enhancer and the +8.9 kb *FOS* enhancer at Day 1 cells might be attributed to an interplay between ETV6 and RUNX factors. In Day 1 cells, high levels of ETV6 may cause the region to function as a repressor, antagonizing the activity of the +8.9 kb enhancer. In contrast, in Day 7 cells, the decreased expression of ETV6 could allow RUNX proteins to switch the repressive function of the enhancer to that of an activator.

The +8.3 kb enhancer is strongly active after CEBPA induction, and we thus expected to find Day 1-specific motifs. Indeed, a PU.1 and a separate ETS2 motif appear at Day 1 cells alongside a CEBPG motif (Fig. EV5I). The presence of dynamic PU.1 and CEBP motifs suggests a functional synergy between the corresponding transcription factors, as in the case of the *IRF8* -69kb enhancer. Other motifs that are present throughout transdifferentiation are also detected by ChromBPnet, such as a MEF2-related binding site.

In conclusion, our analysis provides a possible explanation for the observed antagonistic and stage-specific relationship between the −3.3 kb and 8.8 kb *FOS* PTC enhancers. It also predicts that the −3.3 kb enhancer acts as a repressor because it is bound by the repressive transcription factor ETV6.

## Discussion

Here, we studied putative transcriptional condensates (PTCs) identified by the SEGCOND algorithm (Klonizakis et al, 2023) during an immune cell fate conversion. Unlike other in silico methods, SEGCOND first partitions the genome into large segments and then identifies PTCs through the integration of Hi-C data. Our pipeline revealed enhancer elements far apart across a given chromosome that interact to form large clusters of 3D hubs, regulating gene expression. PTCs were found to be enriched for lineage instructive genes, as has been described for other large enhancer clusters, including LCRs and super-enhancers (Grosveld et al, 1987; Whyte et al, 2013; Parker et al, 2013; Madsen et al, 2020). More specifically, our results showed that individual enhancers within the PTCs of *IRF8* and *FOS*, genes encoding TFs implicated in regulating immune cell development, interact in a differentiation-stage-dependent manner.

PTCs identified in our immune cell transdifferentiation system are associated with highly specialized genes, preferentially expressed in blood cells. Moreover, they are enriched for SNPs that have been linked to autoimmune disorders. For example, the *IRF8* + 83 kb enhancer described here contains an SNP, termed rs9927316, associated with rheumatoid arthritis (Freudenberg et al, 2015). Thus, SEGCOND could represent an attractive framework for the discovery of cell-type-specific disease variants not detected by other algorithms. Moreover, SEGCOND allows the inspection of single nucleotide polymorphisms in a three-dimensional context. This could reveal previously overlooked links between SNPs within regulatory genomic regions and target genes that may be distant in linear space.

Since BRD4 is known to associate with super-enhancers and to form condensates (Sabari et al, 2018), we used its co-localization with PTCs as a proxy for their participation in transcriptional condensates. Accordingly, we found BRD4 to overlap with the PTCs of *IKZF1* and *FOS*, which were also identified by the ROSE algorithm as super-enhancers. Notably, we detected BRD4 foci in the *IRF8* PTC, even though it does not qualify as a super-enhancer in our system. Our results suggest that the ROSE algorithm underestimates the number of large enhancer clusters that overlap with BRD4 hubs in our cell system. Whether PTCs correspond to condensates in live cells is an open question since DNA-FISH requires cell fixation. Therefore, at this stage, SEGCOND can only pinpoint candidate gene loci that overlap with BRD4 assemblies. A surprising aspect of our work was the persistence of BRD4 foci at the *IRF8* PTC following the excision of both the −69 kb and 83 kb enhancers, despite a substantial reduction of *IRF8* gene expression. These observations indicate that recruitment of BRD4 may be necessary but not sufficient for high expression levels of the gene. We therefore speculate that some form of

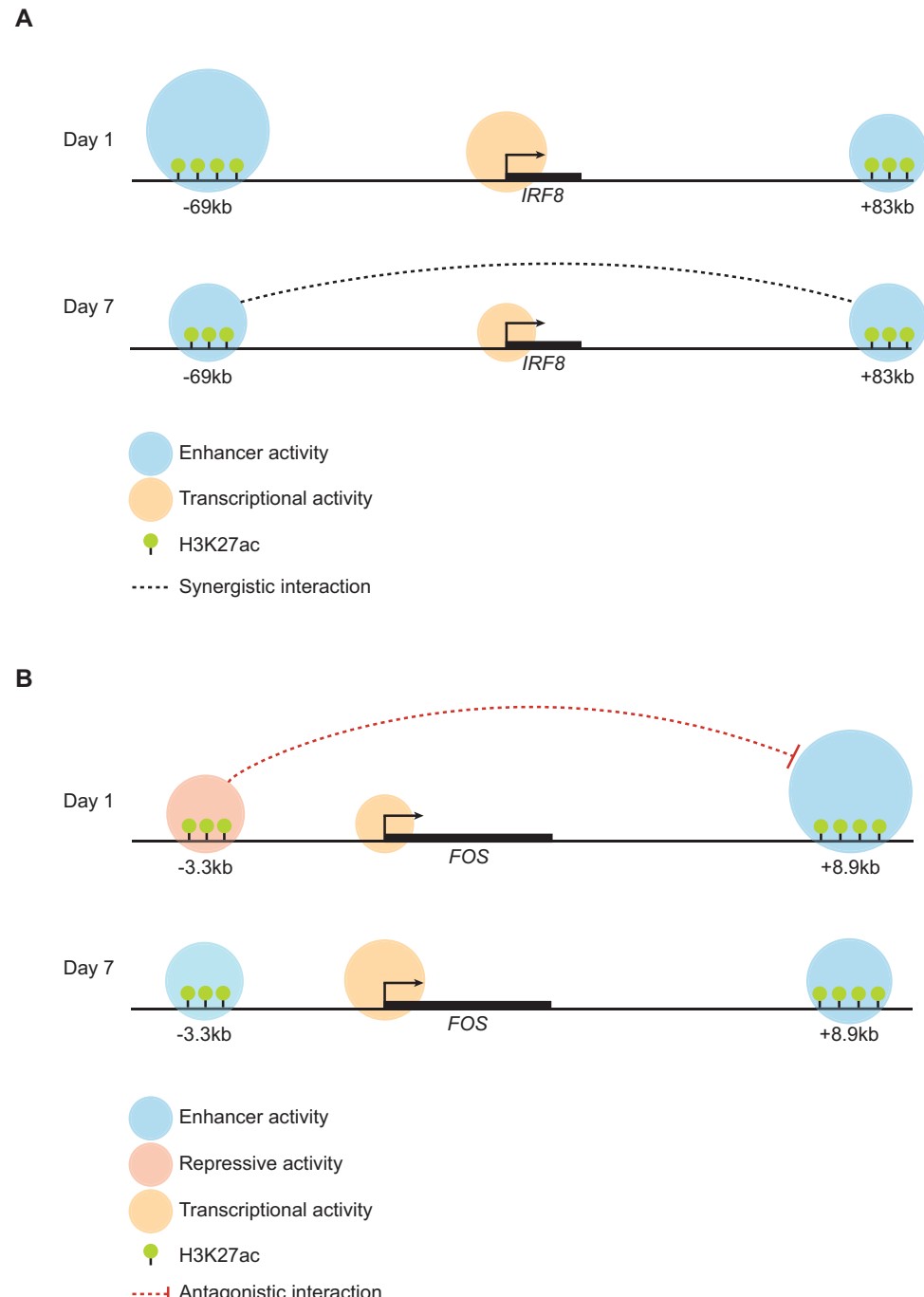

**Figure 6. Differentiation-stage-restricted activities of enhancer pairs during immune cell transdifferentiation.**

(A) Model for enhancer activity within the *IRF8* PTC during transdifferentiation. In Day 1-induced cells, the −69 kb enhancer is stronger than the +83 kb enhancer, leading to high *IRF8* expression levels and a "masking" of +83 kb enhancer activity. The presence of the two enhancers represents a safeguard mechanism, ensuring that *IRF8* expression levels are maintained above a threshold needed for efficient transdifferentiation. In Day 7 cells, the two enhancers display similar activity levels and synergize, driving higher *IRF8* expression levels than expected under an additive model. (B) Model for enhancer activity within the *FOS* PTC during transdifferentiation. In Day 1-induced cells the −3.3 kb enhancer antagonizes the activity of the +8.9 kb enhancer, leading to a decrease in *FOS* expression and a concomitant decrease of transdifferentiation kinetics. This behavior is switched in Day 7 cells, where the two enhancers cooperate independently, driving high *FOS* expression levels.

functional crosstalk between transcription factors recruited at the two *IRF8* enhancers and BRD4 foci is necessary for full activation of *IRF8* expression.

Our knockout data showed that although both the −69 kb and +83 kb *IRF8* enhancers control *IRF8* expression at Day 1 of transdifferentiation,

the −69 kb enhancer is about 2.5-fold stronger than the +83 kb enhancer. Our in silico analysis suggests this is due to the binding of CEBPa and PU.1 in the -69kb enhancer, which does not occur at the +83 kb enhancer. As a result, the activity of the -69kb enhancer overshadows that

of the +83 kb enhancer, "saturating" *IRF8* expression levels. However, in the absence of the −69 kb enhancer, the +83 kb enhancer remains active, representing a mechanism that safeguards *IRF8* expression (Fig. 6A). Therefore, these findings show that the *IRF8* PTC confers robustness to the gene's expression and transdifferentiation, as only the double enhancer excision substantially impaired the two parameters. Such a mechanism may also be relevant to other differentiation processes, where enhancer clusters act as safeguards to maintain expression levels above a critical threshold, as in murine limb development controlled by *Gli3* and *Shox2* (Osterwalder et al, 2018).

The functional characterization of individual enhancers of the *FOS* PTC yielded surprisingly different results. Hence, instead of synergism between its −3.3 kb and +8.9 kb enhancers, it uncovered an antagonistic relationship in Day 1-induced cells, with the −3.3 kb element functioning as a repressor (Fig. 6B). Our study thus reveals that enhancer clusters can be as complex as to contain individual enhancers with opposing functions. ChromBPnet predicts that the interplay between ETV6 and RUNX proteins, bound in the −3.3 kb region, is responsible for this behavior. We speculate that the −3.3 kb region represents an elegant way to limit the maximum rate of mRNA production of the *FOS* gene. Alternating levels of repressor and activator proteins bound in the enhancer could modulate its function and ultimately lock the expression level of the gene in a concentration range required during a particular stage of differentiation. It remains to be tested whether antagonistic interactions between enhancers are relevant for other target genes.

Our results suggest that enhancer clusters can act as transcription factor "control panels", leading to different transcriptional outputs depending on the availability of stage-specific transcription factors and/or co-factors. Accordingly, the relationship between individual enhancers of each cluster proved to be dynamic, with the two examined *IRF8* enhancers synergizing in Day 7-induced cells (corresponding to mature macrophages), but not in Day 1-induced cells (likely corresponding to early myeloid precursors), where their activity levels are independently added up. In contrast, while the two studied *FOS* enhancers also cooperate in Day 7-induced cells, they antagonize each other in Day 1-induced cells. A similar stage-dependent activity has also been reported for the murine *Irf8* enhancer cluster, where Individual enhancers control distinct stages of pDC1 differentiation (Durai et al, 2019) and synergize during a short developmental window (Liu et al, 2023a).

In summary, applying SEGCOND to an immune transdifferentiation system has helped to identify novel enhancer clusters that are missed by other algorithms. Our discovery that individual enhancers within PTCs may not only act synergistically but also antagonistically and in a differentiation-stage-specific manner, could represent as-yet-overlooked new principles of enhancer interactions.

# Methods

### Reagents and tools table

| Reagent/resource | Reference or source | Identifier or catalog number |
|---|---|---|
| **Experimental models** | | |
| BLAER cells | Rapino et al, 2013 | Thomas Graf lab |
| **Recombinant DNA** | | |
| px330-mcherry | Addgene | #98750 |

| Reagent/resource | Reference or source | Identifier or catalog number |
|---|---|---|
| BAC clones | BACPAC genomics | Table 1 |
| **Antibodies** | | |
| Rabbit anti-BRD4 | Abcam | ab128874 |
| Goat anti-rabbit IgG 546 | Thermo Fisher Scientific | A11071 |
| Goat anti-rabbit IgG 488 | Thermo Fisher Scientific | A11070 |
| Fc Receptor Binding Inhibitor Polyclonal Antibody | eBioscience | 16–9161–73 |
| APC-Cy™7 Mouse Anti-Human CD19 | BD Pharmingen | 557791 |
| APC Mouse Anti-Human CD11b | BD Pharmingen | 550019 |
| Rabbit anti-H3K27ac | Diagenode | C15410196 |
| **Oligonucleotides and other sequence-based reagents** | | |
| CRISPR guides | This study | Table 2 |
| PCR screening primers | This study | Table 3 |
| qPCR primers | This study | Table 4 |
| ChIP-qPCR primers | This study | Table 5 |
| **Chemicals, enzymes, and other reagents** | | |
| Cy5-labeled UTP nucleotides | Jena Bioscience | NU-803-CY5-L |
| RPMI 1640 Medium, HEPES | Gibco | 42401018 |
| Fetal bovine serum | Gibco | 10100147 |
| Penicillin/ streptomycin mix | Gibco | 15140122 |
| β-mercaptoethanol | Gibco | 31350010 |
| Phusion High-Fidelity DNA Polymerase | Thermo Fisher | F530L |
| Fluoroshield with DAPI | Sigma-Aldrich | F6057 |
| Formaldehyde | Sigma-Aldrich | F8775-25ML |
| Protein A magnetic beads | Invitrogen | 10002D |
| **Software** | | |
| SEGCOND | Klonizakis et al, 2023 | https://academic.oup.com/bioinformatics/article/39/1/btac742/6832039 |
| gprofiler2 | Kolberg et al, 2020 | https://biit.cs.ut.ee/gprofiler/gost |
| Enrichr | Kuleshov et al, 2016 | https://maayanlab.cloud/Enrichr/ |
| The R Project for Statistical Computing | R Core Team, 2024 | https://www.r-project.org/ |
| EBImage | Pau et al, 2010 | https://www.bioconductor.org/packages/release/bioc/html/EBImage.html |
| Fiji | Schindelin et al, 2012 | https://imagej.net/software/fiji/ |

| Reagent/resource | Reference or source | Identifier or catalog number |
|---|---|---|
| Benchling | https://benchling.com. | N/A |
| ChromBPnet | Pampari et al, 2024 | https://github.com/kundajelab/chrombpnet |
| WashU Epigenome Browser | Li et al, 2022 | https://academic.oup.com/nar/article/50/W1/W774/6567479#google_vignette |
| FlowJo | https://www.bdbiosciences.com/en-eu/products/software/flowjo-v10-software | N/A |
| superEnhancerModelR | Dukler et al, 2017 | https://github.com/CshlSiepelLab/superEnhancerModelR |
| **Other** | | |
| Nick translation kit | Roche | 10976776001 |
| Zymo DNA Clean and Concentrator Kit | Zymo Research | D4013 |
| Poly-L-Lysine-Coated German Glass CoverSlips | Electron Microscopy Sciences | 72292-01 |
| Plasmid Midiprep Kit | Invitrogen | K210004 |
| Cell Line Nucleofector™ Kit C | Lonza | VCA-1004 |
| QuickExtract DNA Extraction Solution | Biosearch Technologies | QE09050 |
| RNeasy Mini Kit | Qiagen | 74104 |
| High-Capacity cDNA Reverse Transcription Kit | Applied Biosystems | 4368814 |
| Click-it Nascent RNA Capture Kit | Thermo Fisher Scientific | C10365 |
| MinElute PCR purification kit | Qiagen | 28004 |

## Methods and protocols

### Application of SEGCOND and datasets used

SEGCOND was applied to datasets derived from an established B-cell-to-macrophage transdifferentiation system (Rapino et al, 2013), as described in (Klonizakis et al, 2023). H3K27ac, H3K4me3, and H3K27me3 ChIP-seq datasets used throughout the study were downloaded from (Borsari et al, 2020). ATAC-seq, RNA-seq, CTCF ChIP-seq, and Hi-C datasets are derived from (Stik et al, 2020), while CEBPA ChIP-seq datasets are from (Choi et al, 2021). PU.1 ChIP-seq data derived from the HL60 cell line was downloaded from GEO (GSM1010843).

### Clustering of genes within PTCs and functional enrichment analysis

To determine which genes are members of PTCs highlighted by SEGCOND, we employed a gene filtering strategy. We only included protein-coding genes that had an average TPM value of >1 across the three timepoints tested. We further asked that genes in enhancer hubs harbored at minimum an H3K27ac peak within a 1000 base pair window from their promoter. Our analysis yielded 4525 genes across all three studied timepoints. We further

separated them into categories according to their time-related PTC membership. GO term analysis was performed in the identified groups by gprofiler2 (Kolberg et al, 2020).

### B-cell and myeloid signature score

Modules of B-cell and myeloid-specific genes were downloaded from a published study (Monaco et al, 2019). We conducted a hypergeometric test in R to determine statistically significant overlaps between genes in PTC association clusters and the isolated modules.

### Separation of genomic categories by SEGCOND

In order to compare the functional characteristics of PTC regions with other meaningful genomic regions, we split the genome into four categories, using SEGCOND. SEGCOND separates the genome into segments and then calculates a log2FC enrichment score and a $P$ value for each segment based on a user-provided characteristic. In our case, the characteristic of interest was the presence of enhancers. Using SEGCOND's quantification schemes for enhancer enrichment within each segment, we separated the genome into four distinct region classes as shown below:

"Inactive": Genomic regions that had a $P$ value of >0.05 and a log2FC < 0

"Moderately active": Genomic regions with a $P$ value > 0.05 and a log2FC >= 0

"Very active": Genomic regions with <=0.05 $P$ value and a log2FC >= 0

"PTC": Genomic regions with a <=0.05 $P$ value, log2FC >= 0 and high Hi-C interaction scores as assessed by SHAMAN (Mendelson Cohen et al, 2017).

### TF-target enrichment scores

To determine whether blood tissue-related transcription factors tend to have more gene targets within PTCs, we utilized the hTFtarget database (Zhang et al, 2020). Transcription factor and target pairs were downloaded and we filtered the table to only pairs known to occur in blood-related tissues. We further filtered the list for transcription factors expressed in our system. We included all protein-coding gene targets, irrespectively from their expression levels, as otherwise we would bias our results towards regions that harbor more active genes. For each transcription factor, we calculated the percentage of total genes regulated by the factor. The "Expected" score of target genes for each TF and each genomic category was thus:

Expected genes = (percentage of gene targets) × (number of genes found within a genomic category).

The final enrichment score per genomic category was calculated as:

Enrichment Score = observed genes/expected genes.

### GTEx tissue-specificity scores

GTEx TPM expression data were downloaded (Lonsdale et al, 2013). For each tissue, genes were ranked according to their expression profile. Expression values were substituted by each gene's rank. In case a tissue had multiple distinct datasets assigned to it (e.g., different brain regions, UV/non-UV exposed skin), gene ranks were averaged. To determine whether a gene is preferentially expressed in Blood tissues, a custom score was generated:

The tissue-specific score for gene I (Blood) = # Tissues with a lower Rank than the Blood rank of gene i / # Tissues.

The range of values of the above score range from 0 to 1. A value of 1 indicates that a gene's rank is the highest in blood tissue and shows biased expression toward it. On the contrary, a value of 0 indicates that a gene's rank is the lowest in blood tissues. The same strategy was applied to get the "Brain" tissue-specificity score. For this analysis, we used all protein-coding genes falling in each SEGCOND genomic category, irrespective of their expression status.

### Enrichment of autoimmune SNPs

Genomic coordinates of SNPs were downloaded from (Maurano et al, 2012) and only autoimmune-related SNPs were kept for further analysis. To determine enrichments of SNPs within different genomic regions, we employed a permutation approach. Using bedtools shuffle (Quinlan and Hall, 2010) coordinates corresponding to the four genomic categories under study were randomly shuffled across the genome a hundred times. After each permutation, overlap with SNPs was calculated and stored. The final enrichment score for each category was:

Enrichment = # Observed SNP Overlap/Average number of SNP overlap with permuted regions.

### Preparation of DNA-FISH probes

To target the *IKZF1*, *IRF8*, and *FOS* PTCs, in-house fluorescent DNA probes were prepared. BAC clones containing the target loci were ordered (BACPAC genomics, Table 1) and a Nick translation

reaction (Roche, Cat 10976776001) was performed where Cy5-labeled UTP nucleotides (Jena, Cat NU-803-CY5-L) were incorporated. The ratio of UTP to non-labeled TTP nucleotides was 30:70. Nick translation was performed following the manufacturer's protocol. After stopping the nick translation reaction, a small quantity of probes was run on a 1% agarose gel, to evaluate the size of produced DNA fragments. Probes that exhibited a smear of 100–500 base pairs were purified using the Zymo DNA Clean and Concentrator Kit (Zymo Research, Cat D4013) and were used for DNA-FISH experiments.

### DNA-FISH coupled with BRD4 immunofluorescence

DNA-FISH was performed as described in (Bolland et al, 2013) with minor modifications. Washes were performed in 24-well plates (Corning, Cat 353047) instead of Coplin jars. Poly-L lysine-coated coverslips were used (Electron Microscopy Sciences, Cat #72292-01) and 200,000 cells were seeded per coverslip. Nitrogen snap-freeze cycles were substituted with dry-ice freeze cycles, with coverslips placed for 15–30 s on top of dry ice per cycle. In total, 50 ng of in-house labeled probes were used per reaction. To perform BRD4 staining using antibodies, the original protocol was extended. Coverslips were re-calibrated in PBST (PBS with 0.2% Tween 20 Merck, Cat 9005-64-5) for 5 min and 50 μl of primary antibody (rabbit anti-BRD4, ref: ab128874, 1:50 dilution) in blocking solution (PBST + 4% BSA) was subsequently placed on top of coverslips. Samples were incubated O/N at 4 °C. The following day, 3 PBST washes of 5 min each were performed followed by incubation of 250 μl secondary antibody solution (Alexa-Fluor 546, goat anti-rabbit, Cat A11071, Alexa-Fluor 488 goat anti-rabbit Cat A11070 for IKZF1 Day 0 samples, 1:500 dilution in PBST + 4% BSA) for 1 h at room temperature. Three rounds of PBST washes were repeated, following mounting with a drop of Fluoroshield with DAPI (Sigma, F6057-20ml). Samples were visualized using an SP5 Inverted Leica confocal

### Table 1.  BAC clone information.

| BAC name | Genomic coordinates (hg38) | Target PTC | Size of BAC |
|---|---|---|---|
| RP11-663L2 | chr7: 50,148,247–50,318,707 | *IKZF1* | 170,460 bp |
| RP11-478M13 | chr16: 85,788,817–85,948,928 | *IRF8* | 160,111 bp |
| CH17-55N12 | chr14: 75,048,853–75,285,587 | *FOS* | 236,734 bp |

### Table 2.  CRISPR sgRNA guides.

| Target region (hg38) | mCherry sgRNA guides | TagBFP sgRNA guides |
|---|---|---|
| *IRF8* + 83 kb, chr16:85,982,380 –85,982,912 | caccGCAAAGACAATGAGAAGCGG, aaacCCGCTTCTCATTGTCTTTGC | caccGTGTGGCCTCTCGTGTCAGT, aaacACTGACACGAGAGGCCACAC |
| *IRF8* -69kb, chr16:85,829,569 –85,830,323 | caccgTGGTGCCCAAGCGTGCCCGG, aaacCCGGGCACGCTTGGGCACCAc | caccGAGTCCAGCCTTCAAATCTG, aaacCAGATTTGAAGGCTGGACTC |
| *FOS* -3.3 kb, chr14:75,274,258 –75,275,500 | caccgTGTATAAAGAACACCCCAG, aaacCTGGGGTGTTCTTTATACAc | caccgTAAAAAGTGGAGCTCACACA, aaacTGTGTGAGCTCCACTTTTTAc |
| *FOS* + 8.9 kb, chr14:75,287,748–75,287,973 | caccgATAGGGTACATTGAATCCTG, aaacCAGGATTCAATGTACCCTATc | caccGAGCTGATGGCCATAAGGCC, aaacGGCCTTATGGCCATCAGCTC |

Bold bases in uppercase represent the CRISPR guide sequence that matches the genomic region to be excised. Non-bold bases in lowercase are linker sequences, needed to clone the CRISPR guides into the px330-derivative vectors.

### Table 3.  PCR screening primers.

| Cell line | Forward primer | Reverse primer |
|---|---|---|
| *IRF8* + 83 kb | CCTGGAGCAGTGATGGACTC | AGACCTCCTTGCAGAACAGC |
| *IRF8* -69kb | GAAGTGGTTCCATCCGCCT | AACATCACTCCAGAGAGCCCA |
| *IRF8* DKO (Derivative of *IRF8* −69 kb KO clone) | CCTGGAGCAGTGATGGACTC | AGACCTCCTTGCAGAACAGC |
| *IRF8* DKO (Derivative of *IRF8* + 83 kb KO clone) | GAAGTGGTTCCATCCGCCT | AACATCACTCCAGAGAGCCCA |
| *FOS* −3.3 kb | TGTGAGTCCCACAGGAATTG | CAATCGAGCTTACAGGGTAGC |
| *FOS* + 8.9 kb | ATTGTCTGTCTCTTATCCCTGAACT | TTGGACCACTCTGCTAAATTGGAT |
| *FOS* DKO (Derivative of *FOS* + 8.9 kb KO clone) | TGTGAGTCCCACAGGAATTG | CAATCGAGCTTACAGGGTAGC |

**Table 4.  qPCR primers.**

| Target | Forward primer | Reverse primer |
|---|---|---|
| IRF8 | AGGAGCCTTCTGTGGACGAT | ACCATCTGGGAGAATGCTGA |
| COX4I1 | AGTGGCGGCAGAATGTTGG | TCCTTCAATGCCTTCTGGCT |
| EMC8 | AGAAGCAGAAGCCGCGTAAG | AACCTGGTTTGGACTGGCAT |
| FOS | CTGTCAACGCGCAGGACTT | GCAGTGACCGTGGGAATGAA |
| NEK9 | TCAACTCGGACTTTGGGAGC | CCAGTGAGTCATCCTCGGTG |
| TMED10 | TGCGTGATACCAACGAGTCAA | CTGGCTGAGGTACAAGGTGG |
| GUSB | CACCAGGATCCACCTCTGAT | TCCAAATGAGCTCTCCAACC |

microscope. For further details on the microscopy parameters used, please refer to the deposited images in the BioImage archive under accession number S-BIAD1480.

### Computational analysis of images from DNA-FISH & BRD4 IF experiments

Images were processed in R using the EBImage package (Pau et al, 2010). Confocal image stacks of $1024 \times 1024$ pixels were processed in the following order:

1. Identification of nuclei. A smoothing Gaussian brush was passed at each DAPI stack (brush size: 31, sigma of 5). Stacks were otsu thresholded and the signal was binarized to identify nuclei. Holes were filled with the fillHull() command.
2. Identification of FISH spots. A smoothing Gaussian brush was passed at each FISH channel stack (brush size: 9, sigma of 5). A stringent thresholding cutoff was picked manually for each set of stacks and was applied to determine DNA-FISH spots. Any spot that did not fall within a nucleus was excluded from the analysis. Additionally, nuclei that did not contain any FISH spots were not considered for downstream analyses.
3. Identification of FISH spot centers. The computeFeatures.moment() command was used to extract centers of identified FISH spots per stack, based on the shape of FISH spots. Centers were then shifted to get the brightest pixel as the central spot. An identifier was assigned to each FISH spot, to track the same FISH spot spanning multiple stacks and only count it once when calculating BRD4 signal overlaps.
4. 3D reconstruction of nuclei and random allocation of spots. Nuclei that harbored FISH spots were stitched in 3D across stacks to reconstruct their total nuclear volume. In total, 50 centers were randomly assigned at each nucleus to determine the background BRD4 signal.
5. Calculation of BRD4 occupancy around FISH centers and random spots. A window of $11 \times 11$ was centered across each identified FISH center and stack. Values from the BRD4 channel were extracted and assigned to each entry of the $11 \times 11$ matrix. Every time a value is assigned to the $11 \times 11$ window, a check takes place to determine whether the window is within a nuclear volume. In the opposite case, a value of 0 is assigned alongside a flag that excludes these entries from further calculations. Finally, an average matrix is calculated for every unique FISH spot spanning multiple image stacks. The same process was repeated for the randomly allocated centers.
6. Generation of contour plots. Median values across all matrix entries of genuine DNA-FISH spots were extracted and plotted as

a contour plot with the filled.contour() command. A similar strategy was employed for the randomly allocated spots. The lowest color scale value was set to blue to match the values of the randomly allocated centers. Consequently, values that are not blue indicate an enrichment of BRD4 intensity in relation to background spots.
7. Calculation of correlation coefficients. In a later step, we wanted to establish whether the BRD4 signal correlates with the increasing intensity of the DNA-FISH signal. To do so, all BRD4 values within the centered $11 \times 11$ windows were extracted and matched to the corresponding FISH-intensity values from the FISH channels. FISH values were binned in 5% percentiles and BRD4 values were assigned to each matching bin. The median values of all BRD4 data points per bin were extracted and plotted.

### Cell culture and transdifferentiation experiments

BLAER cells and enhancer KO derivatives were cultured in RPMI medium (Gibco, catalog no. 42401018) supplemented with 10% fetal bovine serum (Gibco, catalog no. 10100147), 1% glutamine (Gibco, catalog no. 25030081), 1% penicillin/streptomycin mix (Thermo Fisher Scientific, catalog no. 15140122) and 50 μM β-mercaptoethanol (Gibco, catalog no. 31350010). To transdifferentiate BLAER cells and derivatives, 250,000 cells were seeded in 12-well plates in 1 mL of growth medium supplemented by 100 nM E2 (b-estradiol), hIL-3, and hCSF-1 (10 ng ml$^{-1}$). Cells were tested every month for mycoplasma infection and were always found negative.

### Generation of enhancer knockout cell lines

To generate enhancer KO cell lines, we cloned the px330_mCherry and px330_TagBFP plasmids by inserting mCherry and TagBFP sequences (that lack a BbsI cut site), into a base px330 plasmid (Addgene #98750). The base px330 plasmid was a gift from Jinsong Li (Addgene plasmid # 98750; http://n2t.net/addgene:98750; RRID: Addgene_98750). The final plasmids encode the CAS9 enzyme and the corresponding fluorescent marker while they contain a cloning site for an sgRNA of choice. To excise an enhancer, we simultaneously targeted two sites flanking it. We thus cloned sgRNA pairs (Integrated DNA Technologies), after annealing them, in our px330 vectors, generating a set of two px330 plasmids per enhancer excision (Table 2).

For transfection, plasmids were prepared using an Invitrogen Plasmid Midiprep Kit (Invitrogen, K210004). Transfection of BLAER cells was carried out by nucleofection (Amaxa Nucleofector, Lonza) using Kit C (Lonza, VCA-1004) and program X-001,

**Table 5. ChIP-qPCR primers.**

| Target | Forward primer | Reverse primer |
|---|---|---|
| *IRF8* -45.7 kb | ACTAACAGAGGGTTGCCACG | GCCCAGAGTCCATCAAACCA |
| *IRF8* + 30.4 kb | CACCAGCGTAGCTTACGACA | GGGAGAAAACGGGAGTAGGC |
| *IRF8* + 37.2 kb | GGCAGCGGTTAGGGATTTCT | GCCCAAGAACATGATGGACG |
| *FOS* −1 kb | GTCGAGGTATTCCGCCCAG | GTGGTCAGTTCGGGATGACA |
| *FOS* + 8.9 kb | GAATCAGTGCAAAGTCTCTGAAGG | CTCTGTGTCCTCTTAAGCAACTG |
| *ACTB* promoter | TTGCCGACTTCAGAGCAACT | TTGTAGCCTTCATCACGGGC |
| Negative control | TCCCTGCTGGTTCGACTACT | CTGCCAGCAAAAAGCCATGT |

following the manufacturer's protocol. For each cell line, simultaneous transfection of px330_mCherry and px330_TagBFP was performed in the above combinations. We used 3 million cells per nucleofection reaction alongside 3 μg of each plasmid. Cells were cultured for 2 days after nucleofection. Alive cells that were mCherry+ and TagBFP+ were then sorted on a FACSAria II flow cytometer (BD Biosciences). Single cells were seeded in 96-well plates and were left to grow for 2 weeks.

Following 2 weeks of incubation, visible colonies were transferred to a new 96-well plate. A mirror 96-well plate was then established in order to extract genomic DNA from each well and determine which wells harbor cells with the desired excisions. We centrifuged the mirror plates, removed excess medium, and added 50 μl of QuickExtract DNA Extraction Solution (Biosearch Technologies, QE09050) per well. Cell lysates were transferred to PCR 96-well plates and samples were heated in a thermal cycler at 65 °C for 10 min, followed by heating at 98 °C for 5 min. Final lysates were diluted by 100 μl of ddH20 per well. In all, 4 μl of each well was used for a PCR reaction.

### PCR screening
We performed PCR reactions using a Phusion High-Fidelity DNA Polymerase (Thermo Fisher, F530L) to screen for potential homozygous enhancer deletions. PCR primers (Integrated DNA technologies) for each reaction were designed to be at least 100 bp upstream and downstream of predicted CAS9 cuts. The primer sequences are described in Table 3.

### Flow cytometry
To monitor transdifferentiation, we stained surface markers CD19 and MAC1 and performed flow cytometry. Briefly, cells were collected, washed with PBS, and spun down. Cells were resuspended in a blocking solution for 10 min at room temperature using a human FcR binding inhibitor (1:10 dilution, eBiosciences, catalog no. 16–9161–73). Cells were subsequently stained with antibodies against CD19 (1:33 dilution, APC-Cy7 mouse anti-human CD19, BD Pharmingen, catalog no. 557791) and MAC1 (1:10 dilution, APC mouse anti-human CD11b/Mac1, BD Pharmingen, catalog no. 550019) at 4 °C for 20 min in the dark. Cells were afterward washed with PBS and resuspended in PBS with DAPI. FACS analysis was carried out in an LSRII instrument (BD Biosciences). Collected data was analyzed using FlowJo software.

### RNA extraction, reverse transcription, and qPCR
RNA was extracted with RNeasy Mini Kit (Qiagen, 74104) following the manufacturer's protocol. Extracted RNA was

quantified using a NanoDrop spectrophotometer. In total, 500 ng of extracted RNA was used as input for a reverse transcription reaction using the High-Capacity cDNA Reverse Transcription Kit (Applied Biosystems, 4368814). cDNA was then used for quantitative PCR with SYBR Green QPCR Master Mix (Applied Biosystems, A25742). Three technical replicates per gene and condition were used. The primers, spanning exon junctions, used to determine gene expression levels of *IRF8, COX4I1, EMC8, FOS, NEK9, TMED10,* and *GUSB* (housekeeping gene) are depicted in Table 4.

### Nascent RNA capture
Levels of produced *IRF8* mRNA after CEBPA induction were quantified using the Click-iT™ Nascent RNA Capture Kit (Thermo Fisher, C10365). Briefly, cells were resuspended in transdifferentiation medium containing 0.2 mM Ethylene uridine (EU) ribonucleotide homologs and were maintained in culture for 1 day. Cells were then harvested and total RNA was isolated with the RNeasy Mini Kit (Qiagen, 74104) following the manufacturer's protocol. Nascent RNA was then subsequently isolated following the Nascent RNA Capture Kit's instructions. Nascent RNA was immediately used in an RT reaction as described in the "RNA extraction, reverse transcription and qPCR" section.

### Gene expression modeling
Expression values were normalized to the average expression of WT cells within the same transdifferentiation batch. Normalized expression values were used to fit generalized linear models described by Dukler et al, 2017. A brief explanation of the models can be found below:

Additive model: Enhancers control the expression of a gene in an independent, linear fashion. Gene expression can be modeled as:

$$\text{Expression} = b_0 + b_1^* x_1 + b_2^* x_2 + \varepsilon$$

With $x_1$ and $x_2$ being binary variables describing whether an enhancer is present or excised, $b_0$ an intercept term, $b_1$, and $b_2$ coefficients describing enhancer activity and $\varepsilon$ an error term capturing biological and technical noise.

Exponential/Synergistic model: Enhancers control expression multiplicatively, as expected if enhancers display synergistic behavior. Gene expression is modeled as:

$$\text{Expression} = e^{b_0 + b_1^* x_1 + b_2^* x_2} + \varepsilon$$

Logistic model: Transcription of a gene is associated with a low-energy state, while a baseline high-energy state inhibits transcription. Enhancer activity modulates the fraction of time the gene is in

its low-energy state:

$$Expression = g/(1 + e^{-(b0+b1^*x1+b2^*x2)}) + \varepsilon$$

The term g represents the maximum expression level of a gene.

Model coefficients were fit in R using the superEnhancerModelR package (Dukler et al, 2017).

### H3K27ac ChIP and qPCR

Five million BLAER cells were induced and collected after 1 day. Cells were crosslinked for 10 min under rotation by adding formaldehyde (Sigma-Aldrich, F8775-25ML) to a final 1% concentration. The crosslinking reaction was then quenched by adding glycine to a final concentration of 0.125 M and incubating samples for 5 min. Cells were then pelleted, resuspended in ice-cold PBS, and snap-frozen in liquid nitrogen. Pellets were resuspended in ice-cold SDS lysis buffer (1% SDS, 10 mM EDTA, 50 mM Tris pH 8, protease inhibitors) for 15 min and transferred to Bioruptor sonication tubes (diagenode, C30010011). Samples were sonicated using Bioruptor Pico (18 cycles, 30" ON/30" OFF). Samples were then centrifuged, and the supernatant was transferred to a low-bind tube with 900 µl of ChIP dilution buffer (0.01% SDS, 1.1% Triton X-100, 1.2 mM EDTA, 16.7 mM Tris-HCl pH 8.0, 167 mM NaCl, protease inhibitors). 10% of the sample volume was stored as input. In all, 5 µg of chromatin per reaction was incubated overnight at 4 °C with 42 µl of protein A magnetic beads (Invitrogen, 10002D) coupled with H3K27ac antibodies (diagenode, C15410196, 3 µg of antibody per reaction). Beads were collected and washed with Low Salt Buffer (0.1% SDS, 1%, Triton X-100, 2 mM EDTA, 20 mM Tris-HCl pH 8.0, 150 mM NaCl), High Salt Buffer (0.1% SDS, 1%, Triton X-100, 2 mM EDTA, 20 mM Tris-HCl pH 8.0, 500 mM NaCl), LiCl Wash buffer (10 mM Tris-HCl pH 8.0, 1 mM EDTA, 250 mM LiCl, 0.5% NP-40, 0.5% Na-DOC) and TE buffer (10 mM Tris-HCl pH 8.0, 1 mM EDTA). ChIP and input samples were resuspended in Elution buffer (10 mM Tris-HCl pH 8.0, 5 mM EDTA, 300 mM NaCl, 0.5% SDS) and incubated with 40 µg of Proteinase K for 1 h at 55 C. Samples were then incubated overnight at 65 °C to de-crosslink. DNA was finally isolated using a Qiagen MinElute column (Qiagen, 28004).

qPCR was performed to quantify H3K27ac occupancy at regions of interest as described above. In total, 80 pg of ChIP and input DNA was used per sample and reaction. To quantify the H3K27ac signal per loci, we calculated Percent Input scores using the Constant Amount method which is described in (Solomon et al, 2021). Percent Input scores were further normalized to the H3K27ac Input Score of the *ACTB* promoter. The used primers are described in Table 5.

### ChromBPnet analysis

ChromBPnet analysis was performed as indicated in the Githhub tutorial: https://github.com/kundajelab/chrombpnet/wiki. Briefly, ATAC-seq files per timepoint were merged, and only chromosomes 1–22 & X were kept for the analysis. Peak calling was performed in a loose way as recommended by the ChromBPnet user guide with MACS2 and the following command: callpeak -g hs -p 0.01 --shift 75 --extsize 150 --nomodel -B --call-summits. Peaks that were proximal (<=1057 bp) or overlapping hg38 ENCODE blacklisted regions were removed. The same chromosome fold was used for all timepoints (test: chromosomes 1, 3, 6; validation: chromosomes 8,

20; train: all other chromosomes). A bias ChromBPnet model was trained on Day 0 ATAC-seq data and was used to generate the final ChromBPnet models of all timepoints. The bias model "learned" the TN5 motif, but no transcription factor motifs, while the final ChromBPnet models showed the opposite trend. Contribution scores were calculated for all the called accessible peaks falling within the excised regions.

### Statistical analysis

Statistical analysis was carried out using the R programming language. *P* values were used to assess statistically significant differences between groups. A cutoff of 0.05 was used to determine significance. The exact statistical test used and whether the test was paired or unpaired is specified and described in each figure legend. Exact *P* values for each comparison are provided in the Figure source data files. Boxplots depicted in this study display the 25% (lower hinge), 50%, and 75% (upper hinge) percentiles. The upper whisker extends to the largest value provided it is not larger than: 75% percentile + 1.5*(75–25% percentiles). Similarly, the lower whisker extends to the smaller value provided it is not smaller than: 25% percentile – 1.5*(75–25% percentiles).

## Data availability

DNA-FISH and BRD4 immunostaining microscopy images and the R script used for their analysis are deposited in the BioImage archive under the accession number S-BIAD1480 and can be accessed through this link: https://www.ebi.ac.uk/biostudies/bioimages/studies/S-BIAD1480.

The source data of this paper are collected in the following database record: biostudies:S-SCDT-10_1038-S44318-025-00380-w.

## Peer review information

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

## Acknowledgements

The authors would like to thank the Graf and Nikolaou lab members for helpful discussions about the project, as well as Sergi Aranda and Sergi Cuartero for reading the manuscript before submission. The authors also thank members of the Flow Cytometry and Advanced Light Microscopy core facilities. This study was supported by CRG internal funds and the Spanish Ministry of Economy, Industry and Competitiveness (MEIC) Plan Estatal 2019 with project reference number PID2019-109354GB-I00.

## Author contributions

**Antonios Klonizakis**: Conceptualization; Data curation; Software; Formal analysis; Validation; Investigation; Visualization; Methodology; Writing—original draft; Writing—review and editing. **Marc Alcoverro-Bertran**: Validation; Investigation. **Pere Massó**: Validation; Investigation. **Joanna Thomas**: Validation; Investigation. **Luisa de Andrés-Aguayo**: Investigation; Methodology. **Xiao Wei**: Validation; Investigation. **Vassiliki Varamogianni-Mamatsi**: Formal analysis. **Christoforos Nikolaou**: Conceptualization; Supervision; Writing—original draft; Writing—review and editing. **Thomas Graf**: Conceptualization; Supervision; Funding acquisition; Writing—original draft; Writing—review and editing.

Source data underlying figure panels in this paper may have individual authorship assigned. Where available, figure panel/source data authorship is listed in the following database record: biostudies:S-SCDT-10_1038-S44318-025-00380-w.

## Disclosure and competing interests statement

The authors declare no competing interests.

# Expanded View Figures

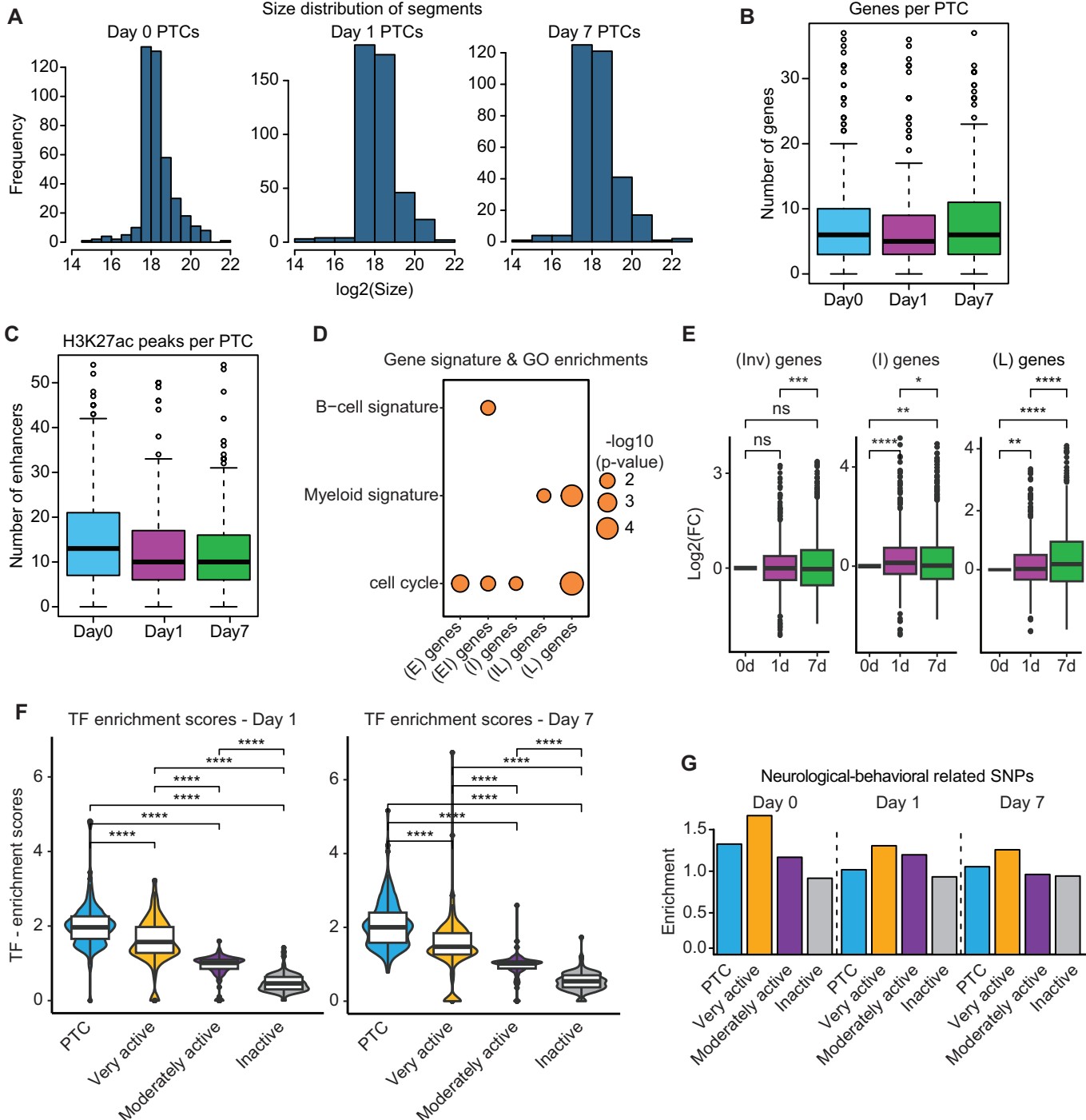

◀ **Figure EV1.  PTCs contain highly expressed, lineage-specific genes across all transdifferentiation stages.**

(A) Size distribution of segments within PTCs. Histograms were used to depict the size distribution of segments across the three transdifferentiation stages. $N = 415$ (Day 0), 437 (Day 1), 316 (Day 7). (B) Distribution of genes per PTC. Genes were filtered as in Fig. 1B. Genes found within different segments of the same PTC were pooled together. $N = 333$ (Day 0), 373 (Day 1), and 271 (Day 7). (C) Distribution of H3K27ac peaks per PTC. H3K27ac peaks of different segments that belong to the same PTC were pooled together. $N = 333$ (Day 0), 373 (Day 1), and 271 (Day 7). (D) Cell signature enrichments of genes within PTC clusters. A custom list of B-cell-specific and myeloid-specific genes was downloaded from (Monaco et al, 2019). Overlap significance between the generated gene lists and the genes within each PTC cluster was assessed via a hypergeometric distribution test. Gene ontology (GO) enrichment analysis for the genes of each PTC group was performed using gprofiler2 (Kolberg et al, 2020) with the "cell cycle" enriched GO term being depicted. The $p$ value of the "cell cycle" GO term was calculated by gprofiler2. $P$ values are represented as circles, with the circle area being proportional to the $P$ value. (E) Expression dynamics of genes falling in the "Invariant", "Intermediate" and "Late" PTC clusters. Values were processed as in Fig. 1D. The paired Wilcoxon signed-rank test was used to determine statistically significant differences (ns $P$ value > 0.05; *$P$ value ≤0.05; **$P$ value ≤0.01; ***$P$ value ≤0.001; ****$P$ value ≤0.0001). $N = 1446$ (Inv genes), 573 (I genes), 680 (L genes). (F) Transcription factor–target enrichment scores for the four SEGCOND genomic groups at Days 1 and 7 of transdifferentiation. Scores were calculated as in Fig. 1F. Statistically significant differences were determined via a paired Wilcoxon rank-sum test (****$P$ value ≤ 0.0001). $N = 131$ for all genomic region types and timepoints. (G) Overlap enrichment of SEGCOND genomic categories with neurological-associated single nucleotide polymorphisms (SNPs) as in Fig. 1H. No significant overlap was observed with this SNP category. Source data are available online for this figure.

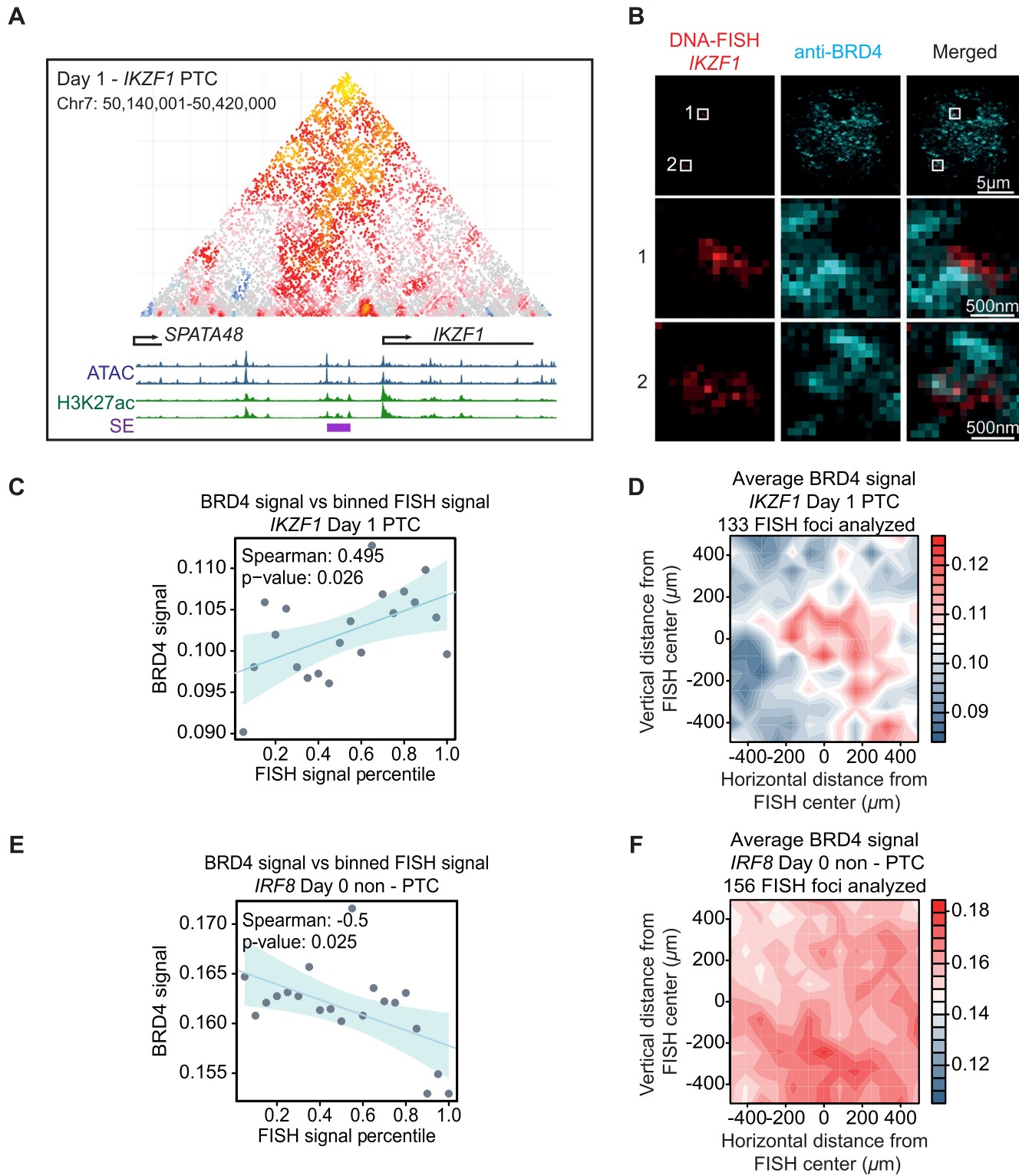

**Figure EV2.  BRD4 hubs associate with active PTCs.**

(A) Overview of the *IKZF1* PTC at Day 1 cells. As in Fig. 2A. (B) DNA-FISH coupled with BRD4 immunofluorescence, targeting the *IKZF1* PTC at Day 1 cells. The third and second rows of images correspond to the zoomed regions falling within the first and second highlighted areas (white frames). (C) DNA-FISH and BRD4 signal correlation analysis for Day-1 *IKZF1* PTC data. Same as in Fig. 2C. (D) Contour plots of BRD4 signal enrichment over Day 1 *IKZF1* DNA-FISH centers. Same as in Fig. 2D. (E) DNA-FISH and BRD4 signal correlation analysis for Day 0 *IRF8* PTC data. Same as in Fig. 2C. Higher DNA-FISH values anti-correlate with high BRD4 values, indicating that BRD4 does not associate with the *IRF8* PTC at Day 0 cells. (F) Contour plots of BRD4 signal enrichment over *IRF8* Day 0 DNA-FISH centers. Same as in Fig. 2D. No enrichment of BRD4 signal at DNA-FISH centers, marking the *IRF8* PTC, can be observed at Day 0 cells.

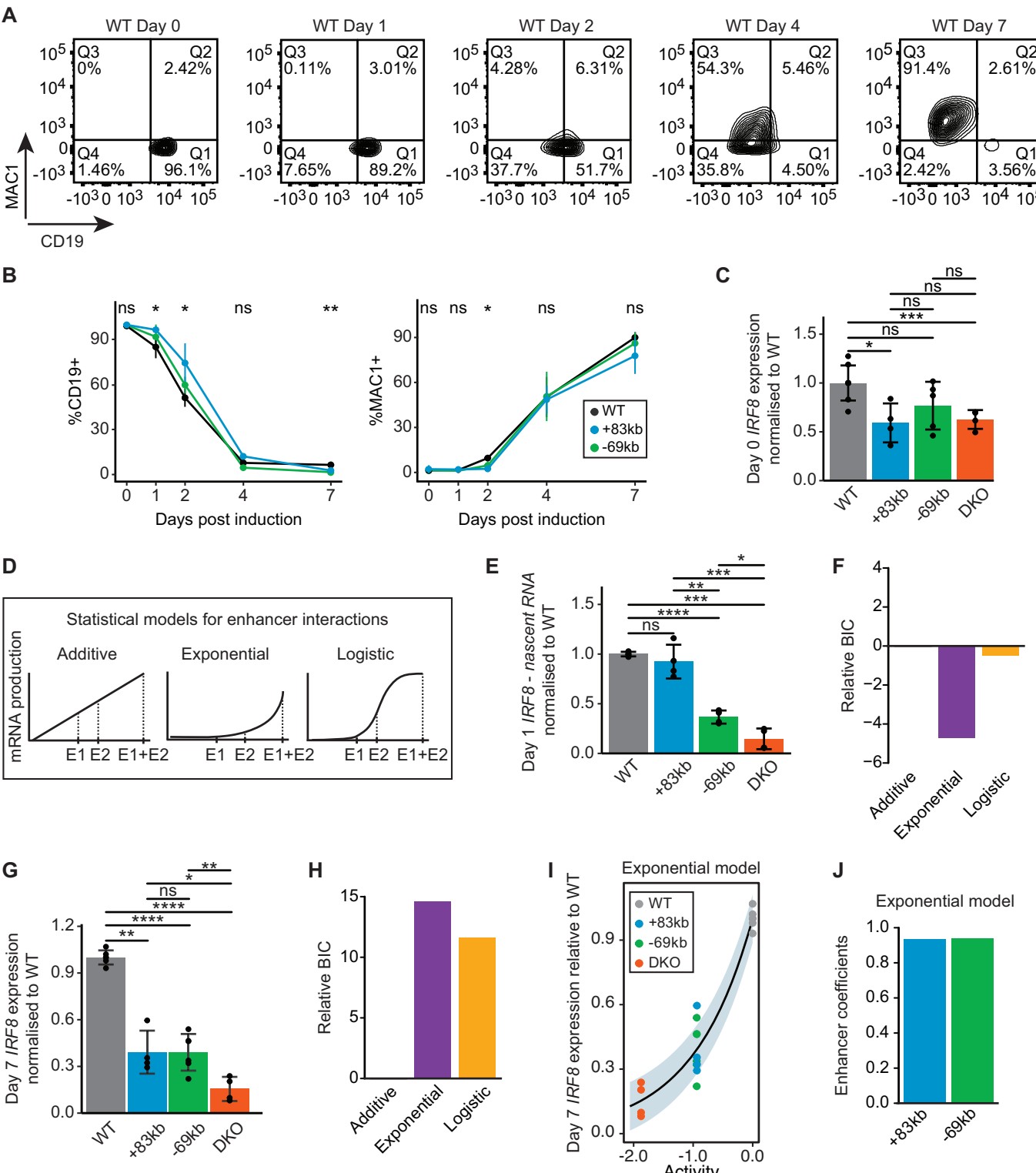

◀ **Figure EV3.   The -69kb and + 83 kb *IRF8* enhancers synergize to drive high *IRF8* expression levels.**

(A) FACS gating strategy used to determine CD19+ and MAC1+ cells during transdifferentiation. Cells falling within Q1 & Q2 were deemed as CD19 + , whereas cells within Q3 & Q4 as MAC1 + . (B) Transdifferentiation kinetics of *IRF8* -69kb KO and +83KO cells compared to BLAER cells, monitored via FACS. The mean ± s.d. is depicted for each timepoint. Statistical significance was determined per timepoint using a two-way ANOVA test, including the transdifferentiation batch as a covariate (ns $P$ value > 0.05; *$P$ value ≤ 0.05; **$P$ value ≤0.01). $N = 3$ (WT, −69 kb, +83 kb Days 0, 1, 2, and 7), and 2 (WT, −69 kb, +83 kb, Day 4) biological replicates. (C) Day 0 *IRF8* expression levels in WT and KO cells. *IRF8* expression before CEBPA induction was quantified via RT-qPCR and normalized against *GUSB* expression. To account for batch-to-batch variations, the expression values from each batch were normalized to the mean WT expression values within the same batch. The mean ± s.d. is depicted, with individual measurements included as points. Statistical significance was determined using a Student's *t* test (ns $P$ value > 0.05; *$P$ value ≤0.05; ***$P$ value ≤0.001). $N = 8$ (WT), 5 (−69 kb), and 4 ( + 83 kb, DKO) biological replicates. (D) Overview of statistical models adapted from (Dukler et al, 2017). In brief, the additive model assumes that each enhancer independently adds its activity, linearly increasing gene expression levels. The exponential model assumes synergy between enhancers, leading to exponential increases in gene expression levels. Finally. the logistic model predicts that transcription occurs in a low-energy state, with each enhancer independently reducing the energy threshold required to reach it. (E) Quantification of newly synthesized *IRF8* mRNA after CEBPA induction. Simultaneous to CEBPA activation, cells were fed an ethylene uridine (EU) ribonucleotide homolog (Click-iT™ Nascent RNA Capture Kit, #C10365), which is incorporated into newly synthesized mRNA molecules and allows their isolation. Isolated nascent RNA was used in RT-qPCR experiments to determine *IRF8* nascent RNA levels in WT and KO cells. Nascent RNA levels of *IRF8* were normalized as in Fig. 3E. The mean ± s.d. is depicted, alongside individual measurements as points. Statistical significance was determined using a Student's *t* test (ns $P$ value > 0.05; *$P$ value ≤0.05; **$P$ value ≤0.01; ***$P$ value ≤0.001; ****$P$ value ≤0.0001). $N = 4$ biological replicates for all samples. (F) Evaluation of best statistical model fit for Day 1 *IRF8* nascent RNA data. As in Fig. 3F. Both the additive and logistic models were found to be good fits. (G) Day 7 *IRF8* expression levels of all generated KO lines and WT cells. Same as Fig. 3E. Statistical significance was determined using a Student's *t* test (ns $P$ value > 0.05; *$P$ value <= 0.05; **$P$ value ≤0.01; ****$P$ value ≤0.0001). $N = 6$ (WT, −69 kb), and 4 ( + 83 kb, DKO) biological replicates. (H) Evaluation of best statistical model fit for Day 7 *IRF8* expression data. As in Fig. 3F, except for the exponential model being the best fit for the dataset. (I) Exponential model fit to *IRF8* Day 7 expression data. Same as in Fig. 3G. (J) Exponential model enhancer coefficients for *IRF8* Day 7 expression data. As in Fig. 3H, except that the model assigns very similar activities to the two enhancers. Source data are available online for this figure.

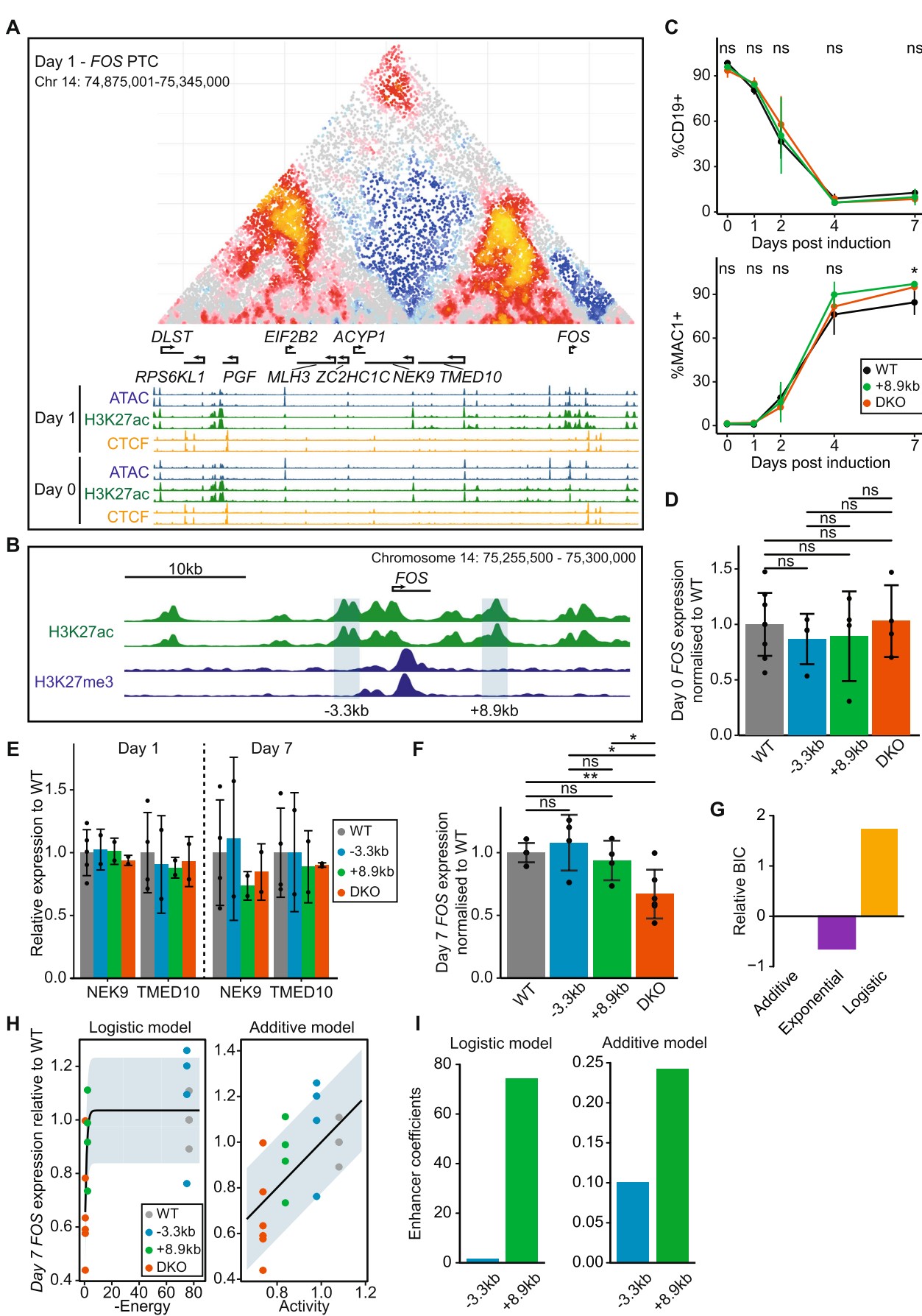

◀

**Figure EV4. The -3.3 kb and + 8.9 kb *FOS* enhancers cooperate at Day 7 cells, safeguarding *FOS* expression.**

(**A**) Overview of the *FOS* PTC on Day 1 of transdifferentiation. As in Fig. 2A. CTCF ChIP-seq tracks are also displayed. (**B**) Overview of the H3K27ac and H3K27me3 marks in the *FOS* PTC at Day 1 cells. Two H3K27ac and two H3K27me3 ChIP-seq tracks of two biological replicates are depicted. The −3.3 kb enhancer isn't decorated with the repressive histone mark H3K27me3. (**C**) Transdifferentiation kinetics of *FOS* + 8.9 kb KO and DKO cells compared to BLAER cells, monitored via FACS. The mean ± s.d. of CD19+ and MAC1+ cells is shown. Statistical significance per timepoint was determined using a two-way ANOVA test, factoring in transdifferentiation batches (ns $P$ value > 0.05; *$P$ value ≤0.05). $N = 4$ (WT Days 0, 1, 2, DKO all timepoints), 3 (WT Days 4, 7, +8.9 kb Days 0, 1, 2), and 2 ( + 8.9 kb Days 4, 7) biological replicates. (**D**) Day 0 *FOS* expression levels in WT and KO cells. Normalization was performed as in Fig. 4G. Statistical significance was determined using a Student's $t$ test (ns $P$ value > 0.05). $N = 9$ (WT), and 4 (−3.3 kb, +8.9 kb, DKO) biological replicates. (**E**) Day 1 and Day 7 expression levels of other genes within the *FOS* PTC. *NEK9* and *TMED10* expression levels were quantified via RT-qPCR and normalized against GUSB expression. The normalized expression values were then divided by the average expression of WT cells. The mean ± s.d. are depicted for each gene, sample, and timepoint. Individual measurements are also depicted as points. Statistical significance of expression differences between cell lines was determined using the Student's $t$ test, and no significant hits were found. $N = 5$ (WT, NEK9 Day 1), 4 (WT, NEK9 Day 7, TMED10 Days 1 and 7), and 2 (all other samples) biological replicates. (**F**) Day 7 *FOS* expression levels of KO and WT cells. As in Fig. 4G. Statistical significance was determined using a Student's $t$ test (ns $P$ value > 0.05; *$P$ value ≤0.05; **$P$ value ≤0.01) $N = 6$ (DKO), 5 (WT), and 4 (−3.3 kb & +8.9 kb) biological replicates. (**G**) Evaluation of best model fit for Day 7 *FOS* expression data. As in Fig. 3F. Both the logistic and additive models were deemed valid. (**H**) Logistic and additive model fits to *FOS* Day 7 expression data. Same as in Fig. 3G. (**I**) Logistic and additive model enhancer coefficients for Day 7 *FOS* expression data. The +8.9 kb enhancer activity is proposed to be higher than the activity of the −3.3 kb enhancer. Source data are available online for this figure.

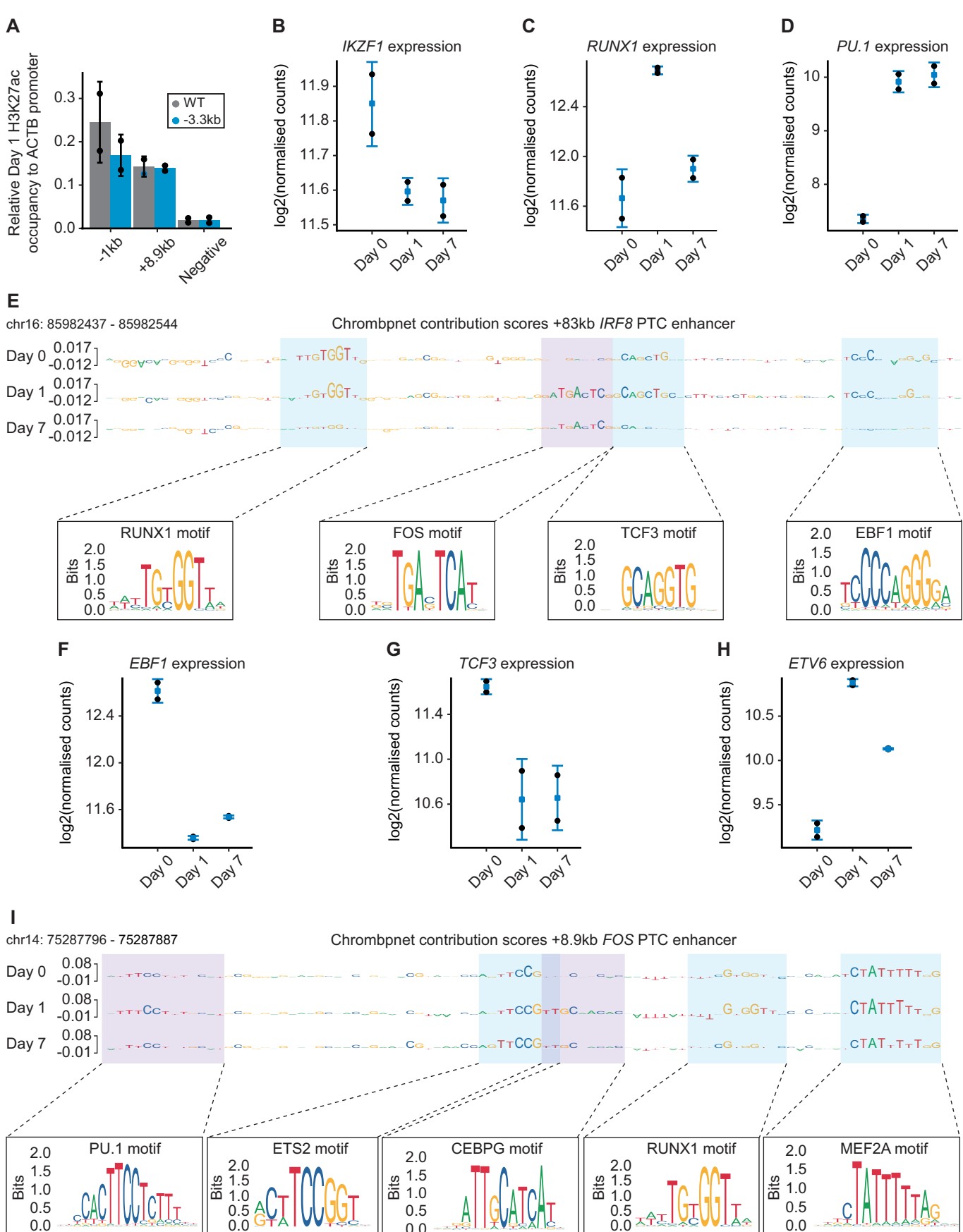

◀ **Figure EV5. In silico analysis of *IRF8* and *FOS* PTC enhancer properties.**

(A) H3K27ac ChIP-qPCR of *FOS* PTC enhancers at Day 1 of transdifferentiation in WT and *FOS* -3.3 kb cells. Primers targeting the +8.9 kb enhancer and a region 1 kb upstream of the *FOS* transcription start site were used. Data was analyzed as in Fig. 5A. $N = 2$ biological replicates for each region and cellular background. (B) Expression profile of *IKZF1* across transdifferentiation. Log2 transformed, DESeq2 variance stabilized counts of two independent RNA-seq replicates are depicted. The mean ± s.d. is depicted in blue, while individual measurements are labeled as points in black. $N = 2$ biological replicates for each timepoint. (C) Expression profile of *RUNX1* across transdifferentiation. As in Fig. EV5B. (D) Expression profile of *PU.1* across transdifferentiation. As in Fig. EV5B. (E) ChromBPnet contribution scores in the +83 kb *IRF8* PTC enhancer at all transdifferentiation stages. As in Fig. 5E. (F) Expression profile of *EBF1* across transdifferentiation. As in Fig. EV5B. (G) Expression profile of *TCF3* across transdifferentiation. As in Fig. EV5B. (H) Expression profile of *ETV6* across transdifferentiation. As in Fig. EV5B. (I) ChromBPnet contribution scores in the +8.9 kb *FOS* PTC enhancer at all transdifferentiation stages. As in Fig. 5E. Source data are available online for this figure.

