## [Peer Review File · The EMBO Journal]

Synergistic and antagonistic activities of IRF8 and FOS enhancer pairs during an immune-cell fate switch

Antonios Klonizakis, Marc Alcoverro-Bertran, Pere Massó, Joanna Thomas, Luisa de Andrés-Aguayo, Xiao Wei, Vassiliki Varamogianni-Mamatsi, Christoforos Nikolaou, and Thomas Graf

Corresponding authors: Thomas Graf (thomas.graf@crg.eu), Christoforos Nikolaou (cnikolaou@fleming.gr)

Review Timeline:

Submission Date:	16th May 24
Editorial Decision:	6th Jul 24
Revision Received:	26th Nov 24
Editorial Decision:	20th Dec 24
Editorial Correspondence:	7th Jan 25
Revision Received:	10th Jan 25
Accepted:	17th Jan 25

Editor: Daniel Klimmeck

Transaction Report:

Dear Dr Thomas Graf,

Thank you for submitting your manuscript for consideration by the EMBO Journal, as well as for your patience with our feedback at this time of the year. Please accept my apologies for the unusual delay with the assessment of your work due to protracted referee input and detailed discussions in the editorial team. Your work has now been seen by three referees with expertise in cell fate control and transcription, whose comments are shown below.

Given the overall interest stated and broader angle of your findings, we are able to invite you to revise your manuscript experimentally to address the referees' comments. I need to stress though that we do require strong support from the referees on a revised version of the study in order to move on to publication of the work.

I would appreciate if you could contact me during the next weeks for exchange e.g. a video call to discuss your perspective on the comments and potential plan for revisions.

Please feel free to contact me if you have any questions or need further input on the referee comments.

When submitting your revised manuscript, please carefully review the instructions below.

Please feel free to approach me any time should you have additional questions related to this.

Thank you for the opportunity to consider your work for publication.

I look forward to your revision.

Best regards,

Daniel Klimmeck

Daniel Klimmeck, PhD
Senior Editor
The EMBO Journal

Instruction for the preparation of your revised manuscript:

- 1) a .docx formatted version of the manuscript text (including legends for main figures, EV figures and tables). Please make sure that the changes are highlighted to be clearly visible.
- 2) individual production quality figure files as .eps, .tif, .jpg (one file per figure).
- 3) a .docx formatted letter INCLUDING the reviewers' reports and your detailed point-by-point response to their comments. As part of the EMBO Press transparent editorial process, the point-by-point response is part of the Review Process File (RPF), which will be published alongside your paper.
- 4) a complete author checklist, which you can download from our author guidelines ([https://wol-prod-cdn.literatumonline.com/pb-assets/embo-site/Author Checklist%20-%20EMBO%20J-1561436015657.xlsx](https://wol-prod-cdn.literatumonline.com/pb-assets/embo-site/Author%20Checklist%20-%20EMBO%20J-1561436015657.xlsx)). Please insert information in the checklist that is also reflected in the manuscript. The completed author checklist will also be part of the RPF.
- 5) Please note that all corresponding authors are required to supply an ORCID ID for their name upon submission of a revised manuscript.
- 6) It is mandatory to include a 'Data Availability' section after the Materials and Methods. Before submitting your revision, primary datasets produced in this study need to be deposited in an appropriate public database, and the accession numbers and database listed under 'Data Availability'. Please remember to provide a reviewer password if the datasets are not yet public (see

<https://www.embopress.org/page/journal/14602075/authorguide#datadeposition>).

7) Our journal encourages inclusion of *data citations in the reference list* to directly cite datasets that were re-used and obtained from public databases. Data citations in the article text are distinct from normal bibliographical citations and should directly link to the database records from which the data can be accessed. In the main text, data citations are formatted as follows: "Data ref: Smith et al, 2001" or "Data ref: NCBI Sequence Read Archive PRJNA342805, 2017". In the Reference list, data citations must be labeled with "[DATASET]". A data reference must provide the database name, accession number/identifiers and a resolvable link to the landing page from which the data can be accessed at the end of the reference. Further instructions are available at .

8) At EMBO Press we ask authors to provide source data for the main and EV figures. Our source data coordinator will contact you to discuss which figure panels we would need source data for and will also provide you with helpful tips on how to upload and organize the files.

Numerical data can be provided as individual .xls or .csv files (including a tab describing the data). For 'blots' or microscopy, uncropped images should be submitted (using a zip archive or a single pdf per main figure if multiple images need to be supplied for one panel). Additional information on source data and instruction on how to label the files are available at .

9) We replaced Supplementary Information with Expanded View (EV) Figures and Tables that are collapsible/expandable online (see examples in <https://www.embopress.org/doi/10.15252/emj.201695874>). A maximum of 5 EV Figures can be typeset. EV Figures should be cited as 'Figure EV1, Figure EV2' etc. in the text and their respective legends should be included in the main text after the legends of regular figures.

11) For data quantification: please specify the name of the statistical test used to generate error bars and P values, the number (n) of independent experiments (specify technical or biological replicates) underlying each data point and the test used to calculate p-values in each figure legend. The figure legends should contain a basic description of n, P and the test applied. Graphs must include a description of the bars and the error bars (s.d., s.e.m.).

We realize that it is difficult to revise to a specific deadline. In the interest of protecting the conceptual advance provided by the work, we recommend a revision within 3 months (4th Oct 2024). Please discuss the revision progress ahead of this time with the editor if you require more time to complete the revisions.

Referee #1:

This manuscript from Klonizakis et al reveals how individual enhancers might synergize or antagonize each other's within large enhancer clusters regulating the expression of lineage-instructing genes upon B cell to macrophage transdifferentiation. Using their previously developed algorithm SEGCOND, the authors defined Putative Transcriptional Condensate (PTCs) as enhancer hubs enriched for highly expressed genes associated with transcriptional condensates, including BRD4. They perform single and/or combinatorial CRISPR/Cas9 mediated deletions of 2 PTCs, harboring myeloid-related IRF8 and Fos genes. These functional experiments document locus-specific and stage-specific enhancer synergies and competition. This study addresses interesting questions regarding the mechanisms of transcription regulation by enhancer hubs and their inter-relationships.

Comments:

-A better description of the PTC features is required, as the authors should not assume that all readers would be familiar with the SEGCOND algorithm. Specifically, they should report for each stage (or PTC category early, later etc) the number of detected PTCs, range of sizes, median numbers of genes and enhancers per PTC. The latter would be particularly relevant for appreciating the degree of synergy or compensation that might occur within such clusters.

-Although significant, many of the reported transcriptional changes within early or late PTCs are rather moderate, and often to the opposite direction than anticipated. It would be helpful for the authors to add a small paragraph discussing potential confounding factors of their SEGCOND approach, which might increase noise.

-The TF enrichment analysis is somewhat confusing and elusive. Wouldn't be better to indicate the specific TF motifs that are enriched in each category? The authors propose that PTCs might function as TF control panels, solely based on this rather confusing TF enrichment analysis. A better explanation and justification is needed.

-Similarly, the authors propose that SEGCOND predicted PTC might support the formation of BRD4 condensates based on their ImmunoFISH analysis. Could they check to what degree this association is perturbed in their individual or DKO clones? Alternatively, and would acute BRD4 inhibition during differentiation (0 to 1 days) have a preferential effect on the PTC-associated genes?

-Another unclear point is how did the authors choose the specific enhancers for each locus. For example, for IRF8 there is a downstream proximal enhancer that has very similar features to the tested ones (acquired H3K27ac signal at day 1 and CEBP binding). It would be beneficial to either go deeper and characterize fully all candidate enhancers with specific features and/or at least try to extract or speculate on the potential features (motifs, accessibility, ABC score etc) that could explain the different behavior of each enhancer. In the case of FOS the selection of very proximal enhancers is also more puzzling, especially given that the authors emphasize the focus on large PTC clusters. The authors should present the entire FOS PTC regions and provide some explanation of how they decided to only test these proximal as opposed to distal enhancers is needed.

-For each enhancer deletion the authors should also test expression of additional genes within or outside the IRF8 or FOS PTC. Is there any gene coordination within PTCs?

-One surprising aspect is the potential direct link between *Irf8* expression and CD19 downregulation (but no *Mac1* change) for the -69kb enhancer. This seems to contradict the case of +83kb, where *Irf8* expression remains unchanged. However, delayed B cell silencing is still observed without affecting the macrophage program, suggesting some indirect effects. The authors should discuss how do they explain impact only on the one but not the other program.

Minor:

-It would be useful to include in the supplementary figures, 1-2 examples per genomic category (HiC and histone marks around random PTC vs Very active regions), so the reader gets a more clear idea of the comparisons

- Similarly to autoimmunity, are PTCs associated with oncogenes and/or genes overexpressed in blood cancer (leukemia, lymphoma)?

- Can the authors directly test the degree to which direct downregulation or upregulation of IRF8 and FOS genes (by shRNA, ectopic expression or literature), recapitulate the phenotypes detected in their enhancer deletions?

Referee #2:

Klonizakis et al. explore the enhancer landscape in the context of CEBPA-induced B-cell to macrophage transdifferentiation. They identify Putative Transcriptional Condensates (PTCs) at 3 different time points and characterize the genomic regions and target genes associated with them. They verify that BRD4 colocalizes with PTCs in a stage specific manner using imaging of the IKZF1 and IRF8 loci. Finally, they use CRISPR to KO individual promoter-distal regulatory elements in the IRF8 and FOS loci to study how individual elements contribute to the regulation of their respective target genes.

This study addresses important questions in gene regulation - most notably, despite our ability to map active regulatory elements in the genome, we still lack an understanding of how regulatory elements work together to modulate target gene expression. Their data showing the impact of knocking out key regulatory elements in the IRF8 and FOS loci are very interesting and add to a growing body of literature highlighting the complex nature of regulatory element interactions. Other data collected in the manuscript appears to be high quality, but these results are ultimately more 'expected' (i.e. that BRD4 would colocalize with transcriptionally active regions, that PTC genes are more likely to be cell type specific, etc.) and are largely confirmatory of previous studies. If the authors want to increase the significance of the work, it might be recommended to investigate additional regulatory elements in PTCs and/or look deeper into the mechanism(s) responsible for the complex regulatory interactions occurring between individual elements that make up the PTCs.

Specific comments:

The most interesting data presented in the manuscript are the results of the regulatory element deletions (Fig. 3B and 4B). Can the authors provide any insight into why some of the elements seem to have a negative influence on nearby gene expression? The consideration of different models (exponential, logistic, etc.) is a welcome approach here, but with so few data points (and substantial error bars) it is hard to place too much emphasis/trust in those conclusions.

In the first paragraph of the results, it's unclear if the identification of PTCs is being conducted on previously published data and/or if the PTCs were previously reported already. This should be made clear to the reader so that they know where to find the appropriate information that supports those conclusions.

Fig 1d and Supp Fig. 1b are difficult to make out - it seems that some of them favor up-regulation, while others are meant to show decreasing levels, but as far as I can make out it just looks like the overall variance increases. Consider visualizing these results differently, maybe showing how individual genes are regulated (or their connections, e.g. heatmap or something like that).

In Fig. 1F and Supp. Fig. 1C on TF enrichment, what does that mean, how should it be interpreted? Is this meant to imply that some genes are regulated by TFs and others are not? Is the fact that PTC genes have more TF enrichment just a function of the fact that they have more regulatory elements in their vicinity relative to the other categories?

Referee #3:

Klonizakis and colleagues employ their novel computational tool (SEGCOND) along with their cell line model of trans-differentiation (BLAER cells) to examine the role of two specific enhancer pairs (for the FOS and IRF8 genes). They also provide some experimental evidence for the condensate-associated protein BRD4 to colocalize with the genomic regions of interest (termed PTCs). Overall, although the use of the novel computational tool seems appropriate, the experiments are well designed and the observations are intriguing, there remains many unanswered questions and additional details that are required to be disclosed before publication. I do consider that the findings reported upon in this manuscript will be of interest to the EMBO readership and will likely be of general scientific significance once some modifications have been made. Specific details are mentioned below.

Specific comments:

The genes which are associated with the PTCs (identified from SEGCOND) are not listed in the manuscript. These gene lists should be included. Additionally, could genes outside of the PTC (but nearby) also be potentially affected by the PTC? If the gene associations are relinked but with a less stringent limitation of residing within a PTC (but by also including a specific number of bases nearby as well) how does this change the number of genes identified??

The authors employ the human target TF database to analyze their PTC classification. More information is required to allow the authors to understand this analysis. The authors need to provide the filtered list of TFs which they use in the downstream analysis. The authors could also provide the enrichment score for each TF (in an ordered manner i.e. waterfall plot) particularly to support the authors conclusion that there are a higher variety of transcription factors involved with PTC-associated genes. Please explain the results and conclusions from this section in more detail and clarity.

The colocalization studies identify occurrence of BRD4 at the specified enhancer regions using DNA FISH and computation analysis. However, additional controls/experiments are required to be confident in the conclusions from this component of the study. A) Does an enhancer from a non-PTC region exhibit this same co-localization? B) BRD4 is only one potential marker of condensates... can the authors show a similar pattern with another marker of condensates? C) What does the IKZF1 PTC look like at Day 7, or at a time when it should be non-active? D) Does this interaction occur in non-cell line models? (i.e. primary cells). E) Does the same interaction occur at the FOS enhancer locus? minor points: How many cells were imaged and used in each imaging plot/analysis. Is the label of the SE in plot 2A in the correct place? And could you display where the FISH probe is in relation to the gene in plot 2A (or if it is the entire region, specify this more clearly)?

The knockout experiments examining the two enhancers provide the most intriguing results, however they raise potentially more questions than they answer. In particular, it would be important to know what effects the KO of the enhancers has on the surrounding regions. This could include ATAC seq, Histone marks etc. This could help determine whether these deletions are breaking down the whole structure of the area or whether it is the specific DNA segment/TF motifs of the enhancer which are leading to these observations. Please attempt to address this experimentally in one of the enhancer examples.

In regards to the IRF8 enhancers, the authors need to discuss why the +83kb enhancer displayed no early difference in IRF8 expression yet it had an the most notable effect on the trans-differentiation out of the two single knockouts. Also this differentiation data should be included with the DKO data in the main figure (move and merge from Supplement).

Although potentially out of the scope of this paper, does RNA knockdown of FOS/IRF8 in these BlaER cells lead to similar changes in trans-differentiation kinetics? This could help explain whether it is the resultant change in expression of the TF or whether there are other complexities arising from the deletion of the enhancer DNA sequences.

Response to reviewer's comments

EMBOJ-2024-117895: "Synergistic and antagonistic activities of *IRF8* and *FOS* enhancer pairs during an immune cell fate switch"

Please find enclosed our revised manuscript, with significant changes marked in blue. We would like to thank all the reviewers for their helpful and fair comments which we believe substantially improved our work. We have now delved more into the mechanism behind the behaviors of the excised enhancers, as all reviewers agreed that this was the most interesting aspect of our study. We have introduced several changes and added new data. The main changes are listed below:

1. In the revised manuscript we have performed H3K27ac CHIP-qPCR experiments to assess whether the excised enhancers affect the chromatin structure of PTCs. We also provide a new and detailed *in silico* analysis based on the ChromBPnet algorithm to estimate the bound transcription factors in the excised loci. We have added a new Figure and a new EV Figure to incorporate our findings.
2. Although none of the reviewers requested it, we now include the data of a 2nd clone harboring a +83kb enhancer deletion to make our conclusions more robust.
3. We now provide the raw files and source data for the figure panels.

Please find below a point-by-point response to all the reviewer's comments. Original comments are in black, while our responses are in blue.

Referee#1:

This manuscript from Klonizakis et al reveals how individual enhancers might synergize or antagonize each other's within large enhancer clusters regulating the expression of lineage-instructing genes upon B cell to macrophage transdifferentiation. Using their previously developed algorithm SEGCOND, the authors defined Putative Transcriptional Condensate (PTCs) as enhancer hubs enriched for highly expressed genes associated with transcriptional condensates, including BRD4. They perform single and/or combinatorial CRISPR/Cas9 mediated deletions of 2 PTCs, harboring myeloid-related *IRF8* and *Fos* genes. These functional experiments document locus-specific and stage-specific enhancer synergies and competition. This study addresses interesting questions regarding the mechanisms of transcription regulation by enhancer hubs and their inter-relationships.

We would like to thank the reviewer for the detailed summary of our work and for providing several suggestions that significantly strengthened our manuscript.

Comments:

-A better description of the PTC features is required, as the authors should not assume that all readers would be familiar with the SEGCOND algorithm. Specifically, they should report for each stage (or PTC category early, later etc) the number of detected PTCs, range of sizes, median numbers of genes and enhancers per PTC. The latter would be particularly relevant

for appreciating the degree of synergy or compensation that might occur within such clusters.

We have now extended the first paragraph of the Results section with a detailed description of the requested PTC features. We have also added three new panels (Figure EV1, A,B, and C) to address the reviewer's comment.

-Although significant, many of the reported transcriptional changes within early or late PTCs are rather moderate, and often to the opposite direction than anticipated. It would be helpful for the authors to add a small paragraph discussing potential confounding factors of their SEGCOND approach, which might increase noise

Given the large sizes of PTCs, we cannot rule out possible insulation effects within them. This could lead to different gene expression responses throughout transdifferentiation, where one gene gets activated/shut off but the rest of the genes within the same PTC do not. Moreover, promoter-enhancer compatibility may represent another barrier limiting genes' co-regulation within PTCs. However, we would like to stress that the average gene expression changes follow the expected trends and are statistically significant. We have now added the following sentences in the fourth paragraph of the Results section:

“In some instances, we noticed only moderate changes in gene expression (Fig. EV1E). Given the large size of PTCs, we cannot exclude that only a specific set of genes within each PTC is predominantly affected by changes in chromatin structure and three-dimensional chromatin interactions. Thus, CTCF binding and/or promoter–enhancer compatibility could be responsible for local insulation effects within PTCs, restricting expression changes to a subset of genes.”

-The TF enrichment analysis is somewhat confusing and elusive. Wouldn't be better to indicate the specific TF motifs that are enriched in each category? The authors propose that PTCs might function as TF control panels, solely based on this rather confusing TF enrichment analysis. A better explanation and justification is needed.

We have updated the corresponding Figure 1 panel in order to better present the main idea behind our TF enrichment analysis. Additionally, we further clarified our analysis by expanding the second paragraph of the second section of Results:

“To explore this, we identified transcription factors (TFs) that are expressed in our cell system and matched them to their target genes, utilizing information from the hTFtarget database (Zhang et al., 2020). We then classified the target genes of each TF by their localization in the genome and assigned them to one of the genomic categories defined by SEGCOND (Fig. 1F). We next calculated a “background” network that randomly assigned the target genes to genomic categories according to their total sizes. Using this approach, we calculated an enrichment score for each SEGCOND-defined genomic category and each transcription factor, quantifying whether or not target genes of a given TF are overrepresented.”

-Similarly, the authors propose that SEGCOND predicted PTC might support the formation of BRD4 condensates based on their ImmunofISH analysis. Could they check to what degree

this association is perturbed in their individual or DKO clones? Alternatively, and would acute BRD4 inhibition during differentiation (0 to 1 days) have a preferential effect on the PTC-associated genes?

To address these questions we have now performed DNA-FISH and BRD4 immunostaining at Day 1 *IRF8* DKO cells and added our new results in Figures 5B, 5C, and 5D. This revealed that in spite of the sharp drop in *IRF8* expression levels after the double enhancer excision, BRD4 is still present at the *IRF8* PTC. This finding suggests that the excised enhancers are not necessary for BRD4 recruitment. We cannot rule out that other, unperturbed, enhancers in the *IRF8* PTC are responsible for its recruitment. In fact, we assessed H3K27ac deposition in other enhancers within the *IRF8* PTC in *IRF8* DKO cells via ChIP-qPCR (Figure 5A) and found that the enhancers are still decorated with the mark.

-Another unclear point is how did the authors choose the specific enhancers for each locus. For example, for *IRF8* there is a downstream proximal enhancer that has very similar features to the tested ones (acquired H3K27ac signal at day 1 and CEBP binding).

To better explain our rationale in choosing the -69kb and +83kb enhancers we have expanded the first paragraph of the fourth section of Results. The reason why we selected only two enhancers is that establishing enhancer KO clones is quite challenging and testing all possible enhancer combinations in the *IRF8* PTC would have been impractical. We chose the -69kb and +83kb enhancers of the *IRF8* PTC based on increased accessibility and H3K27ac occupancy after CEBPA induction and its unique binding by both CEBPA and PU.1.

It would be beneficial to either go deeper and characterize fully all candidate enhancers with specific features and/or at least try to extract or speculate on the potential features (motifs, accessibility, ABC score etc) that could explain the different behavior of each enhancer.

We attempted to KO an additional enhancer within the *IRF8* PTC to further characterize the PTC's function (-45.7kb enhancer in Figure 5A) with 4 different guide RNA combinations but only got heterozygous cells after screening. We thus discontinued these endeavors. Instead, we turned to an *in silico* approach, based on the ChromBPnet algorithm, to better characterize the enhancer features that explain the observed differences in each excised enhancer's function. ChromBPnet is a deep learning model that trains on ATAC-seq data and can reveal sequences that "contribute" to chromatin accessibility at selected regions. We ran ChromBPnet on Day 0, 1, and 7 ATAC-seq data and focused on the excised enhancer regions. Our analysis revealed sequence motifs that become relevant after CEBPA induction and that could explain the functional differences observed in our study. To describe this, we have added two new Results sections, updated the Discussion, and added new panels (Figure 5D-E and Figure EV5B-I) to explain our findings.

In the case of *FOS* the selection of very proximal enhancers is also more puzzling, especially given that the authors emphasize the focus on large PTC clusters. The authors should present the entire *FOS* PTC regions and provide some explanation of how they decided to only test these proximal as opposed to distal enhancers is needed.

We have now included the complete *FOS* PTC of 470kb in the revised manuscript (Fig. EV4A). It contains multiple genes and enhancers, most of which are already active at Day 0 cells. The two enhancers chosen are uniquely active at Day 1 cells, close to the *FOS* gene, and flanked by CTCF sites (Fig. EV4A). Targeting enhancers within the boundaries of CTCF sites minimizes covariates in our analysis. Finally, the two *FOS* enhancers are also part of a super-enhancer, making our findings more generally appealing. To better explain our rationale for the enhancer choice we have now expanded the first paragraph of the Result section on page 8.

-For each enhancer deletion the authors should also test expression of additional genes within or outside the *IRF8* or *FOS* PTC. Is there any gene coordination within PTCs?

To address this criticism we have now included RT-qPCR experiments with Day 1 and Day 7 cells, targeting the *EMC8* and *COX4I1* genes found within the *IRF8* PTC (Figure 3D). Their expression is not affected by the excision of the two enhancers, probably because of an insulation effect of multiple CTCF sites that flank the *IRF8* gene.

Similarly, we tested the expression of *NEK9* and *TMED10* within the *FOS* PTC (Figure EV4A). These were chosen from among five expressed genes in the locus that show enriched interactions between their TSS and the *FOS* gene. However, the expression of these two genes was not affected by the excised enhancers, suggesting promoter-enhancer selectivity (Fig EV4E).

-One surprising aspect is the potential direct link between *Irf8* expression and CD19 downregulation (but no *Mac1* change) for the -69kb enhancer. This seems to contradict the case of +83kb, where *Irf8* expression remains unchanged. However, delayed B cell silencing is still observed without affecting the macrophage program, suggesting some indirect effects. The authors should discuss how do they explain impact only on the one but not the other program.

We were also puzzled by the transdifferentiation dynamics of *IRF8* +83kb KO cells. To explore this further, we isolated RNA from non-induced *IRF8* KO cell lines and WT cells and observed that *IRF8* levels were already reduced after +83kb KO excision (Fig. EV3C). We believe that this reduction in *IRF8* expression in uninduced cells is the reason for the slight delay in the silencing of CD19 in the +83kb KO cells, especially given that other genes in the *IRF8* PTC are not affected by the enhancer deletion.

Due to the observed differences in the starting levels of *IRF8* among the different KO lines, we also quantified *nascent IRF8* mRNA molecules produced for one day after CEBPA activation. We obtained identical results as with our total RNA measurements, ensuring that our statistical modeling and conclusions regarding Day 1 bulk RNA data are robust (Figures EV3E & EV3F)

Minor:

-It would be useful to include in the supplementary figures, 1-2 examples per genomic category (HiC and histone marks around random PTC vs Very active regions), so the reader gets a more clear idea of the comparisons

We now include two examples per genomic region in the Appendix file.

- Similarly to autoimmunity, are PTCs associated with oncogenes and/or genes overexpressed in blood cancer (leukemia, lymphoma)?

We used Enrichr, and inspected terms related to Human diseases (Human Phenotype Ontology atlas) in order to assess whether genes in PTCs are related to blood cancer. Indeed, the top hit for Day 0 and Day 1 cells was “Lymphoma” (q-values 0.01502 & 0.009357). On the contrary, Day 7 cells were not enriched, as CEBPA-induced macrophages adopt a more quiescent state. We have expanded page 5 of Results to incorporate these findings.

- Can the authors directly test the degree to which direct downregulation or upregulation of IRF8 and FOS genes (by shRNA, ectopic expression or literature), recapitulate the phenotypes detected in their enhancer deletions?

We mainly wanted to focus on the enhancer relationships across transdifferentiation and less on *IRF8* and *FOS* as potential regulators of the process. Nevertheless, both proteins are known to be regulators of the myeloid fate (Cai et al., 2007)(Tamura et al., 2000), and the observed changes in their expression match the “expected” differentiation phenotypes. For example, large drops in *IRF8* expression led to a delay in *MAC1* upregulation (Fig. 3C), while an increase in *FOS* expression led to the opposite effect (Fig 4E). More mild drops in *IRF8* and *FOS* expression do not seem to affect the activation of the myeloid program (Fig. EV3B and Fig. EV4C), hinting at possible expression thresholds that are important for differentiation.

Referee #2:

Klonizakis et al. explore the enhancer landscape in the context of CEBPA-induced B-cell to macrophage transdifferentiation. They identify Putative Transcriptional Condensates (PTCs) at 3 different time points and characterize the genomic regions and target genes associated with them. They verify that BRD4 colocalizes with PTCs in a stage specific manner using imaging of the *IKZF1* and *IRF8* loci. Finally, they use CRISPR to KO individual promoter-distal regulatory elements in the *IRF8* and *FOS* loci to study how individual elements contribute to the regulation of their respective target genes.

This study addresses important questions in gene regulation - most notably, despite our ability to map active regulatory elements in the genome, we still lack an understanding of how regulatory elements work together to modulate target gene expression. Their data showing the impact of knocking out key regulatory elements in the *IRF8* and *FOS* loci are very interesting and add to a growing body of literature highlighting the complex nature of regulatory element interactions. Other data collected in the manuscript appears to be high quality, but these results are ultimately more 'expected' (i.e. that BRD4 would colocalize with transcriptionally active regions, that PTC genes are more likely to be cell type specific, etc.) and are largely confirmatory of previous studies. If the authors want to increase the significance of the work, it might be recommended to investigate additional regulatory

elements in PTCs and/or look deeper into the mechanism(s) responsible for the complex regulatory interactions occurring between individual elements that make up the PTCs.

We would like to thank the reviewer for the positive comments on our work and the recommendation to further investigate the molecular events underlying the function of the excised enhancers.

Specific comments:

The most interesting data presented in the manuscript are the results of the regulatory element deletions (Fig. 3B and 4B). Can the authors provide any insight into why some of the elements seem to have a negative influence on nearby gene expression? The consideration of different models (exponential, logistic, etc.) is a welcome approach here, but with so few data points (and substantial error bars) it is hard to place too much emphasis/trust in those conclusions.

We have now explored potential ways to explain the observed antagonistic interaction in the *FOS* PTC. To this end, we performed ChIP-qPCR experiments in WT and -3.3kb KO cells, targeting a region near the *FOS* promoter and the +8.9kb enhancer. The H3K27ac signal at these loci is not affected (Fig. EV5A), ruling out a mechanism that modulates enhancer activity through the deposition of the histone mark.

We also performed a detailed *in silico* analysis based on the ChromBPnet algorithm. The algorithm is based on a convolutional neural network approach, trained on ATAC-seq data of Day 0, 1, and 7 cells. It allows the identification of DNA sequences within a region that leads to increased chromatin accessibility. Inspecting the *FOS* -3.3kb enhancer, we found two putative sites bound by the repressor ETV6, thus explaining the negative influence of this enhancer. We have included these data in a new Figure (Figure 5) and have expanded the Results, Methods, and Discussion sections to explain our findings. However, the proposed model still needs to be experimentally validated, which is, however, out of the scope of this study. That said, we took extra care not to overstate our findings in the revised manuscript.

In the first paragraph of the results, it's unclear if the identification of PTCs is being conducted on previously published data and/or if the PTCs were previously reported already. This should be made clear to the reader so that they know where to find the appropriate information that supports those conclusions.

We apologize for not stating clearly that the PTC identification was already performed in an earlier study with previously published datasets (Klonizakis et al. 2023). We modified the first paragraph of the Results to explain this.

Fig 1d and Supp Fig. 1b are difficult to make out - it seems that some of them favor up-regulation, while others are meant to show decreasing levels, but as far as I can make out it just looks like the overall variance increases. Consider visualizing these results differently, maybe showing how individual genes are regulated (or their connections, e.g. heatmap or something like that).

We thank the reviewer for the suggestion but opted for maintaining the boxplots in Fig. 1D and Fig. EV1E. While we agree that in specific cases, changes in gene expression are rather subtle (e.g., “Intermediate” genes in Fig. EV1E), in the majority of the cases genes behave as expected. For example, we observed a clear reduction in the expression of genes within “Early” and “Early-Intermediate” PTCs, as documented in Fig. 1D.

In Fig. 1F and Supp. Fig. 1C on TF enrichment, what does that mean, how should it be interpreted? Is this meant to imply that some genes are regulated by TFs and others are not?

To better explain our analysis, we have now expanded the Results section and updated the Fig. 1F panel. Briefly, for our analysis, we prepared a list of transcription factors and their target genes. We then quantified the number of each transcription factor's gene targets found in each category (PTC, very active, moderately active, inactive) and compared this to the calculated “expected” number of target genes per category. Per transcription factor, we then quantified the number of its gene targets found in each category (PTC, very active, moderately active, inactive) and calculated the expected number of target genes in each category. The expected number of targets takes into account the total size of all the genomic regions. For example, “Inactive” regions occupy 74,2% of the genome, while “PTC” regions occupy 5,5%. We then calculated an enrichment score for each transcription factor and each of the 4 SEGCOND categories. Scores greater than 1 indicate that there are more gene targets within the corresponding genomic region than expected, as is the case for PTCs.

Accordingly, we further clarified our analysis by expanding the second paragraph of the second section of Results: *“To explore this, we identified transcription factors (TFs) that are expressed in our cell system and matched them to their target genes, utilizing information from the hTFtarget database (Zhang et al., 2020). We then classified the target genes of each TF by their localization in the genome and assigned them to one of the genomic categories defined by SEGCOND (Fig. 1F). We next calculated a “background” network that randomly assigned the target genes to genomic categories according to their total sizes. Using this approach, we calculated an enrichment score for each SEGCOND-defined genomic category and each transcription factor, quantifying whether or not target genes of a given TF are overrepresented.”*

Is the fact that PTC genes have more TF enrichment just a function of the fact that they have more regulatory elements in their vicinity relative to the other categories?

Our results suggest that a cohort of transcription factors is responsible for the establishment of PTCs rather than a “master” regulator. We cannot rule out that the enrichment scores observed are due to the higher abundance of regulatory elements found within PTCs. However, the “Very Active” category is also highly enriched in regulatory elements but shows smaller TF enrichment scores than PTC regions.

Referee #3:

Klonizakis and colleagues employ their novel computational tool (SEGCOND) along with their cell line model of trans-differentiation (BLAER cells) to examine the role of two specific

enhancer pairs (for the FOS and IRF8 genes). They also provide some experimental evidence for the condensate-associated protein BRD4 to colocalize with the genomic regions of interest (termed PTCs). Overall, although the use of the novel computational tool seems appropriate, the experiments are well designed and the observations are intriguing, there remains many unanswered questions and additional details that are required to be disclosed before publication. I do consider that the findings reported upon in this manuscript will be of interest to the EMBO readership and will likely be of general scientific significance once some modifications have been made. Specific details are mentioned below.

We thank the reviewer for finding our work interesting and providing suggestions on improving it.

Specific comments:

The genes which are associated with the PTCs (identified from SEGCOND) are not listed in the manuscript. These gene lists should be included. Additionally, could genes outside of the PTC (but nearby) also be potentially affected by the PTC? If the gene associations are relinked but with a less stringent limitation of residing within a PTC (but by also including a specific number of bases nearby as well) how does this change the number of genes identified??

We now provide an Excel spreadsheet with the number of filtered genes used in our analysis, showing at which timepoint genes were found to be within a PTC (Source data file). We have now also included a more detailed characterization of our PTCs (Figures EV1A-C). We do not perceive any additional merit in including genes that are close to the PTCs since we found that adding genes that are ≤ 50 kb in the vicinity of PTCs only has a minor impact:

	Filtered genes within PTCs	Filtered genes within/close (≤ 50 kb) to PTCs
Day 0	2897	3030
Day 1	3042	3216
Day 7	2755	2873

The authors employ the human target TF database to analyze their PTC classification. More information is required to allow the authors to understand this analysis. The authors need to provide the filtered list of TFs which they use in the downstream analysis. The authors could also provide the enrichment score for each TF (in an ordered manner i.e. waterfall plot) particularly to support the authors conclusion that there are a higher variety of transcription factors involved with PTC-associated genes. Please explain the results and conclusions from this section in more detail and clarity.

To improve the clarity of the performed analysis we have now expanded the corresponding Figure panel (Fig. 1F) and added the following sentences in the Results section (page 4): *“To explore this, we identified transcription factors (TFs) that are expressed in our cell system and matched them to their target genes, utilizing information from the hTFtarget database (Zhang et al., 2020). We then classified the target genes of each TF by their localization in the genome and assigned them to one of the genomic categories defined by SEGCOND (Fig.*

1F). We next calculated a “background” network that randomly assigned the target genes to genomic categories according to their total sizes. Using this approach, we calculated an enrichment score for each SEGCOND-defined genomic category and each transcription factor, quantifying whether or not target genes of a given TF are overrepresented.”

Also, we have now provided the filtered list of transcription factors and their target genes in an Excel Spreadsheet (Source data file). Our conclusion that a wider variety of transcription factors is involved in regulating PTC-associated genes is supported by the data represented as violin plots in Fig. 1F & Fig. EV1F. Briefly, for our analysis, we prepared a list of transcription factors and their target genes. Per transcription factor, we then quantified the number of its gene targets found in each category (PTC, very active, moderately active, inactive) and calculated the “expected” number of target genes in each category. The “expected” number of targets takes into account the total size of all the genomic regions. For example, “Inactive” regions occupy 74,2% of the genome, while “PTC” regions occupy 5,5%. We then calculated an enrichment score for each transcription factor and each of the 4 SEGCOND categories. Scores greater than 1 indicate that there are more gene targets within the corresponding genomic region than expected. Since PTCs display the highest scores among all categories, they are the most enriched.

The colocalization studies identify occurrence of BRD4 at the specified enhancer regions using DNA FISH and computation analysis. However, additional controls/experiments are required to be confident in the conclusions from this component of the study. 67

The DNA-FISH experiments in our system are challenging as cells grow in suspension and our FISH probes are prepared in-house. We have, therefore, validated only a handful of probes and were able to generate data only for Days 0 and 1 of transdifferentiation. Applying our protocol for Day 7 cells was not possible as the nuclei exhibited a distorted morphology and even after optimizing several steps (e.g. HCl incubation, detergent concentration), we were not able to obtain useful data.

A) Does an enhancer from a non-PTC region exhibit this same co-localization?

We think that this question is beyond the scope of the paper. Our goal was to determine whether BRD4 assembles in PTC regions, not at individual enhancers. Strengthening the confidence in our data, the *IRF8* PTC at Day 0 of transdifferentiation does not co-localize significantly with BRD4 although the gene is expressed in Day 0 cells.

B) BRD4 is only one potential marker of condensates... can the authors show a similar pattern with another marker of condensates?

We have tried to perform DNA-FISH experiments coupled with MED1 staining, as MED1 is another widely recognized component of transcriptional condensates (Sabari et al., 2018). However, these efforts were not fruitful.

C) What does the IKZF1 PTC look like at Day 7, or at a time when it should be non-active?

We initially selected the *IKZF1* PTC because it does not classify as a PTC in Day 7 cells and hoped that we could test whether or not BRD4 overlaps in Day 7 cells. However, as mentioned above, our protocol is unsuitable for induced macrophages.

D) Does this interaction occur in non-cell line models? (i.e. primary cells).

To explore this with primary B cells is technically a lot more challenging than with our cell line system, which was difficult enough.

E) Does the same interaction occur at the *FOS* enhancer locus?

We have now included DNA-FISH and BRD4 immunostaining experiments targeting the *FOS* PTC at Day 1 cells (Figures 4A-C). Our results indicate that indeed, BRD4 foci overlap with the *FOS* PTC as well.

minor points: How many cells were imaged and used in each imaging plot/analysis.

We have now added the total number of DNA FISH foci analyzed in each experiment in the title of the contour plots depicted in Figures 2D, 2H, S2D, S2F, and 4C.

Is the label of the SE in plot 2A in the correct place?

Yes, the called SE by the ROSE algorithm in Day 0 cells is as depicted in Fig. 2A.

And could you display where the FISH probe is in relation to the gene in plot 2A (or if it is the entire region, specify this more clearly)?

We have now updated the corresponding Figure legends to state the size of the BAC clone used for each experiment. We also added more detailed information on the BAC clones used in the Methods section.

The knockout experiments examining the two enhancers provide the most intriguing results, however they raise potentially more questions than they answer. In particular, it would be important to know what effects the KO of the enhancers has on the surrounding regions. This could include ATAC seq, Histone marks etc. This could help determine whether these deletions are breaking down the whole structure of the area or whether it is the specific DNA segment/TF motifs of the enhancer which are leading to these observations. Please attempt to address this experimentally in one of the enhancer examples.

We would like to thank the reviewer for the suggestion to perform ATAC-seq or ChIP-seq experiments to investigate in more depth the mechanisms behind the function of the excised enhancers. We have therefore performed H3K27ac ChIP experiments with Day 1 *IRF8* PTC KO and WT cells and qPCR experiments targeting three H3K27ac decorated regions across the PTC (new Fig. 5A). The results show that enhancer excisions do not affect the overall structure of the PTC, as other H3K27ac decorated regions are unaffected (Fig. 5A).

We also evaluated whether the *FOS* repression mediated by the -3.3kb enhancer on Day 1 involved the disruption of H3K27ac deposition in the *FOS* promoter and +8.9kb enhancer. To this end, we performed H3K27ac ChIP with Day 1 *FOS* -3.3kb and WT cells and targeted the *FOS* +8.9kb enhancer and a region close to the *FOS* transcription start site via qPCR (Fig. EV5A). We once again found that the deletion does not lead to any H3K27ac deposition changes. We added a new Figure (Figure 5) and expanded the Results, Discussion, and Methods sections to incorporate our new findings.

In regards to the *IRF8* enhancers, the authors need to discuss why the +83kb enhancer displayed no early difference in *IRF8* expression yet it had the most notable effect on the trans-differentiation out of the two single knockouts. Also this differentiation data should be included with the DKO data in the main figure (move and merge from Supplement).

We were also surprised by the observed delay in CD19 silencing in the +83kb *IRF8* clone as Day 1 *IRF8* mRNA levels were not significantly altered. We now have a potential explanation. After repeating measurements of *IRF8* expression in non-induced *IRF8* KO clones, we observed that *IRF8* levels are already lower in DKO and +83kb KO cells than in WT cells (Fig. EV3C). Thus, we speculate that the initial difference in *IRF8* expression may be the reason for the delay. We also ruled out the involvement of other genes within the *IRF8* PTC, as their expression remains unperturbed following enhancer excision (Fig. 3D). We have now expanded the Results with an extra paragraph that describes these findings (page 7). We also thank the reviewer for the suggestion of moving the +83kb and -69kb transdifferentiation data into the main figure together with the DKO data. However, we would like to keep them as they are, in order to better highlight the strong effect of double enhancer excision in the transdifferentiation dynamics.

Finally, we were concerned that the difference in starting *IRF8* levels may have affected our Day 1 modeling data and expression measurements. To address this, we repeated the Day 1 qPCR experiments by monitoring nascent *IRF8* RNA for one day after CEBPA induction and performed statistical modeling using this new data (Figures EV3E & EV3F). The results were identical to those obtained for total RNA.

Although potentially out of the scope of this paper, does RNA knockdown of *FOS*/*IRF8* in these BlaER cells lead to similar changes in trans-differentiation kinetics? This could help explain whether it is the resultant change in expression of the TF or whether there are other complexities arising from the deletion of the enhancer DNA sequences.

We have not tested this suggestion as *IRF8* and *FOS* are known regulators involved in myeloid specification (Cai et al., 2007)(Tamura et al., 2000). In addition, changes in their expression correlate with changes in transdifferentiation dynamics. Thus, a sharp reduction in *IRF8* expression leads to a delay in *MAC1* activation (Fig. 3C), while an increase in *FOS* expression caused the reverse phenotype (Fig. 4F). Finally, the expression of genes falling within the *IRF8* and *FOS* PTCs was found not to be affected by the excisions of the enhancers (Fig. 3D & Fig. EV4E), suggesting that the changes in the levels of *IRF8* and *FOS* are drivers of transdifferentiation perturbations.

Dear Dr Thomas Graf,

Thank you for submitting your revised manuscript (EMBOJ-2024-117895R) to The EMBO Journal. Your amended study was sent back to the three referees for their scientific re-evaluation, and we have received detailed comments from two of them, which I enclose below. Please note that while referee #2 got delayed and has not yet provided his/her recommendations, we have editorially assessed your response to the critique raised by this reviewer and found the issues to be addressed satisfactorily. As you will see, the other experts state that the work has been substantially enhanced by the revisions and they are now in favour of publication.

Thus, we are pleased to inform you that your manuscript has been accepted in principle for publication in The EMBO Journal, pending there are no overriding technical issues still raised by referee #2.

I will let you know as soon as we receive his-her pending re-report.

We now need you to take care of a number of issues related to formatting and data presentation as detailed below, which should be addressed at re-submission.

Please contact me at any time if you have additional questions related to below points.

Thank you for giving us the chance to consider your manuscript for The EMBO Journal. I look forward to your final revision.

Again, please contact me at any time if you need any help or have further questions.

Best regards,

Daniel Klimmeck

> Please add up to five keywords to your study.

> Author Contributions: Remove the author contributions information from the manuscript text. Note that CRediT has replaced the traditional author contributions section as of now because it offers a systematic machine-readable author contributions format that allows for more effective research assessment. and use the free text boxes beneath each contributing author's name to add specific details on the author's contribution.

More information is available in our guide to authors.

> Adjust the title of the 'Declaration of Interests' section to 'Disclosure and Competing Interests Statement'.

> Correct order of manuscript sections: Abstract / Keywords / Introduction / Results / Discussion /Methods / Data Availability / Acknowledgements / Disclosure and competing interests statement / /References / Figure legends / Tables and their legends / Expanded View Figure legends

> "Materials and Methods" should be renamed "Methods".

> References: please adjust reference format to EMBO Journal format, 10 authors et al. . DOIs should only be used for preprints and datasets that have not been published yet.

> Data availability section: please move before 'Acknowledgments'.

> Add a separate 'Statistical analysis' section, detailing the algorithms and statistical tests applied.

> Remove the Reagents and Tools table from the manuscript and provide as a separate file using the existing template in the Guide For Authors, listing key reagents, experimental models, software and relevant equipment.

> Source data: 2B, 2F are currently missing - they might be deposited to BioImage under S-BIAD1480, but we need a direct URL and access provided in the DAS; also provided SD for EV figures and additional panels; source data should be uploaded as one (zipped) file per figure.

> Please recheck references for the following bioRxiv entries and update citations if in the meantime

published as regular article: Borsari et al. (2020), Mendelson Cohen et al. (2017), Mota-Gomez et al. (2022).

> Consider additional changes and comments from our production team as indicated below:

- DAS:

1. Please note that the specific URLs for Bioimage dataset S-BIAD1480 is not provided in the data availability statement.

>>> now added AD 4.12.24

2. Please note that reviewer access codes for S-BIAD1480 dataset is not provided in the data availability statement.

>>> now accessible AD 4.12.24

- Figure legends:

1. Please note that the exact p values are not provided in the legends of figures 1D, F, G; 3C, E; 4F, G; EV1 E, F; EV3 B, C, E, G; EV4 F.

2. Please indicate the statistical test used for data analysis in the legends of figures EV1 D.

3. Please note that in figures 1D, F, G; 3C, E; 4F, G; EV1 E, F; EV3 B, C, G; EV4C, D, F; there is a mismatch between the annotated p values in the figure legend and the annotated p values in the figure file that should be corrected.

4. Please note that the box plots need to be defined in terms of minima, maxima, centre, bounds of box and whiskers, and percentile in the legends of figures 1D, F, G; EV1 B, C, E, F.

5. Please note that n=2 in figures 3B, C, D; 4E, 5A; EV3 B; EV4 C, E; EV5 A, B, C, D, F, G, H.

Further information is available in our Guide For Authors:

Please use the link below to submit your revision:

Referee #1:

In the revised version, the authors addressed many of my key concerns with additional explanations, orthogonal analysis and new experiments. Importantly, they clearly state many of the limitations of their approach and they provide fair evaluation of their results and well-supported conclusions. Although some gaps remain and many of the experiments and findings could be strengthened with additional experiments and controls, I believe that the current data and discussion provide a complete and interesting story for publication.

Referee #3:

The authors have comprehensively attempted to address all my concerns. I appreciate the effort to address every comment and believe that the added data/information has greatly improved the paper. I congratulate the authors on their work, it is a very nice addition to the field.

Dear Thomas,

Further to below, I can now share that we still received additional re-comments by referee #2, which are enclosed. You will see that also this expert is supportive of the revised work.

Please consider the remaining arguments of this referee carefully and amend the manuscript by additional experiment or adjusting claims DNA discussion of the results where appropriate.

Let me know any time if there are any points related to be discussed.

Best regards,
Daniel

EMBOJ-2024-117895R Re-review comments Referee #2:

The authors have adequately addressed my concerns in this review. The expanded explanation of the TF analysis (Fig. 1f) is nice improvement, and the inclusion of additional clones provides more robustness to the results.

Two additional comments on the new material:

First, when investigating how element deletion impacts H3K27ac levels in fig. 5a/b, why not measure H3K27ac at the IRF8 promoter as well, for which there is corroborating evidence (i.e. changes in expression Fig. 3b) which can also serve as a true positive (in theory). Similar for the Jun PTC results (Fig. EV5A). This would help solidify the results and conclusions in this part of the manuscript.

Second, the inclusion of the ChromBPnet analysis and discussion helps provide some context (and speculation) about how the regulatory elements might be organized to confer their specific activities, and is generally a good addition to the manuscript. However, there is some concern that the assignment of transcription factor identity to the sequences identified might be inferring too much and is worded too strong in the manuscript. In particular, the assignment of ETS transcription factors (of which I assume there are at least a dozen different ETS TFs expressed in these cells), might be dubious given the high similarity reported in most of their associated DNA motifs. For example, the sequence bound by the first PU.1 site in Fig. 5E and the first sequence bound by ETV6 in Fig. 5F could probably be assigned to either ETV6 or PU.1 (not to mention probably a bunch of other ETS factors). ETS2 and IKZF1 motifs are not too different from the ETV5 and PU.1 motifs either. I would recommend the authors either find additional corroborating evidence to support these assignments (e.g. new or public ChIP data), or at the minimum place a large caveat/disclaimer on their analysis that describes the nuance of these TF assignments that underscores the relative confidence in their speculation.

Dear Dr Thomas Graf,

Thank you for submitting your revised manuscript (EMBOJ-2024-117895R) to The EMBO Journal. Your amended study was sent back to the three referees for their scientific re-evaluation, and we have received detailed comments from two of them, which I enclose below. Please note that while referee #2 got delayed and has not yet provided his/her comments, we have editorially assessed your response to the critique raised by this reviewer and found the issues to be addressed satisfactorily. As you will see, the other experts state that the work has been substantially enhanced by the revisions and they are now in favour of publication.

Thus, we are pleased to inform you that your manuscript has been accepted in principle for publication in The EMBO Journal, pending there are no overriding technical issues still raised by referee #2. I will let you know as soon as we receive his-her pending re-report.

We now need you to take care of a number of issues related to formatting and data presentation as detailed below, which should be addressed at re-submission.

Please contact me at any time if you have additional questions related to below points.

Thank you for giving us the chance to consider your manuscript for The EMBO Journal. I look forward to your final revision.

Again, please contact me at any time if you need any help or have further questions.

Best regards,

Daniel Klimmeck

>> Please add up to five keywords to your study.

>> Author Contributions: Remove the author contributions information from the manuscript text. Note that CRediT has replaced the traditional author contributions section as of now because it offers a systematic machine-readable author contributions format that allows for more effective research assessment. and use the free text boxes beneath each contributing author's name to add specific details on the author's contribution.

More information is available in our guide to authors.
<https://www.embopress.org/page/journal/14602075/authorguide>

>> Adjust the title of the 'Declaration of Interests' section to 'Disclosure and Competing Interests Statement'.

>> Correct order of manuscript sections: Abstract / Keywords / Introduction / Results / Discussion / Methods / Data Availability / Acknowledgements / Disclosure and competing interests statement // References / Figure legends / Tables and their legends / Expanded View Figure legends

>> "Materials and Methods" should be renamed "Methods".

>> References: please adjust reference format to EMBO Journal format, 10 authors et al. . DOIs should only be used for preprints and datasets that have not been published yet.

>> Data availability section: please move before 'Acknowledgments'.

>> Add a separate 'Statistical analysis' section, detailing the algorithms and statistical tests applied.

>> Remove the Reagents and Tools table from the manuscript and provide as a separate file using the existing template in the Guide For Authors, listing key reagents, experimental models, software and relevant equipment.

>> Source data: 2B, 2F are currently missing - they might be deposited to BioImage under S-BIAD1480, but we need a direct URL and access provided in the DAS; also provided SD for EV figures and additional panels; source data should be uploaded as one (zipped) file per figure.

>> Please recheck references for the following bioRxiv entries and update citations if in the meantime published as regular article: Borsari et al. (2020), Mendelson Cohen et al. (2017), Mota-Gomez et al. (2022).

>> Consider additional changes and comments from our production team as indicated below:

- DAS:

1. Please note that the specific URLs for Bioimage dataset S-BIAD1480 is not provided in the data availability statement.

>>> now added AD 4.12.24

2. Please note that reviewer access codes for S-BIAD1480 dataset is not provided in the data availability statement.

>>> now accessible AD 4.12.24

- Figure legends:

1. Please note that the exact p values are not provided in the legends of figures 1D, F, G; 3C, E; 4F, G; EV1 E, F; EV3 B, C, E, G; EV4 F.

2. Please indicate the statistical test used for data analysis in the legends of figures EV1 D.

3. Please note that in figures 1D, F, G; 3C, E; 4F, G; EV1 E, F; EV3 B, C, G; EV4C, D, F; there is a mismatch between the annotated p values in the figure legend and the annotated p values in the figure file that should be corrected.

4. Please note that the box plots need to be defined in terms of minima, maxima, centre, bounds of box and whiskers, and percentile in the legends of figures 1D, F, G; EV1 B, C, E, F.

5. Please note that n=2 in figures 3B, C, D; 4E, 5A; EV3 B; EV4 C, E; EV5 A, B, C, D, F, G, H.

Referee #1:

In the revised version, the authors addressed many of my key concerns with additional explanations, orthogonal analysis and new experiments. Importantly, they clearly state many of the limitations of their approach and they provide fair evaluation of their results and well-supported conclusions. Although some gaps remain and many of the experiments and findings could be strengthened with additional experiments and controls, I believe that there the current data and discussion provide a complete and interesting story for publication.

Referee #3:

The authors have comprehensively attempted to address all my concerns. I appreciate the effort to address every comment and believe that the added data/information has greatly improved the paper. I congratulate the authors on their work, it is a very nice addition to the field.

Referee #1:

In the revised version, the authors addressed many of my key concerns with additional explanations, orthogonal analysis and new experiments. Importantly, they clearly state many of the limitations of their approach and they provide fair evaluation of their results and well-supported conclusions. Although some gaps remain and many of the experiments and findings could be strengthened with additional experiments and controls, I believe that the current data and discussion provide a complete and interesting story for publication.

We would like to thank the reviewer for the positive comments regarding our work and for the previous constructive comments that improved our manuscript.

Referee #2:

The authors have adequately addressed my concerns in this review. The expanded explanation of the TF analysis (Fig. 1f) is nice improvement, and the inclusion of additional clones provides more robustness to the results.

We would like to thank the reviewer for the positive evaluation of our revised manuscript.

Two additional comments on the new material:

First, when investigating how element deletion impacts H3K27ac levels in fig. 5a/b, why not measure H3K27ac at the *IRF8* promoter as well, for which there is corroborating evidence (i.e. changes in expression Fig. 3b) which can also serve as a true positive (in theory). Similar for the Jun PTC results (Fig. EV5A). This would help solidify the results and conclusions in this part of the manuscript.

The *IRF8* promoter could, in theory, serve as an additional control, as the reviewer suggested. However, in the case of the *IRF8* PTC, we were mostly interested in enhancer interactions, and thus sought to evaluate whether there were any changes in H3K27ac deposition in other enhancer elements following -69kb and/or +83kb enhancer excision. With regard to the Jun PTC (we assume the reviewer refers to the *FOS* PTC), we included the promoter region to clarify the mechanism of repression mediated by the -3.3kb element.

Second, the inclusion of the ChromBPnet analysis and discussion helps provide some context (and speculation) about how the regulatory elements might be organized to confer their specific activities, and is generally a good addition to the manuscript. However, there is some concern that the assignment of transcription factor identity to the sequences identified might be inferring too much and is worded too strong in the manuscript. In particular, the assignment of ETS transcription factors (of which I assume there are at least a dozen different ETS TFs expressed in these cells), might be dubious given the high similarity reported in most of their associated DNA motifs. For example, the sequence bound by the first PU.1 site in Fig. 5E and the first sequence bound by ETV6 in Fig. 5F could probably be assigned to either ETV6 or PU.1 (not to mention probably a bunch of other ETS factors). ETS2 and IKZF1 motifs are not too different from the ETV5 and PU.1 motifs either. I would recommend the authors either find additional corroborating evidence to support these

assignments (e.g. new or public CHIP data), or at the minimum place a large caveat/disclaimer on their analysis that describes the nuance of these TF assignments that underscores the relative confidence in their speculation.

We agree that the assignment of defined transcription factors to sequence motifs should be done with caution. Accordingly, we have now modified our manuscript to point out the caveats of our analysis (page 11, highlighted in blue).

Referee #3:

The authors have comprehensively attempted to address all my concerns. I appreciate the effort to address every comment and believe that the added data/information has greatly improved the paper. I congratulate the authors on their work, it is a very nice addition to the field.

We would like to thank the reviewer for the positive comments on our work and for the previous suggestions on how to improve our manuscript.

Dear Dr Graf, dear Dr Nikolaou,

Thank you for submitting the revised version of your manuscript. I have now evaluated your amended manuscript and concluded that the remaining minor concerns have been sufficiently addressed.

I am thus pleased to inform you that your manuscript has been accepted for publication in the EMBO Journal.

On a different note, I would like to alert you that EMBO Press offers a format for a video-synopsis of work published with us, which essentially is a short, author-generated film explaining the core findings in hand drawings, and, as we believe, can be very useful to increase visibility of the work. Please see the following link for representative examples and their integration into the article web page:

<https://www.embopress.org/doi/full/10.15252/emj.2019103932>

Finally, we have noted that the submitted version of your article is also posted on the preprint platform bioRxiv. We would appreciate if you could alert bioRxiv on the acceptance of this manuscript at The EMBO Journal in order to allow for an update of the entry status. Thank you in advance!

Best regards,

Daniel Klimmeck

Daniel Klimmeck, PhD
Senior Editor
The EMBO Journal

EMBO
Postfach 1022-40
Meyerhofstrasse 1
D-69117 Heidelberg
contact@embojournal.org

t